# What makes a sustainability tool valuable, practical and useful in real-world healthcare practice? A mixed-methods study on the development of the Long Term Success Tool in Northwest London

Laura Lennox,[1,2] Cathal Doyle,[3] Julie E Reed,[3,4] Derek Bell

[1]Department of Medicine, CLAHRC for Northwest London, London, UK
[2]Department of Medicine, Imperial College London, London, UK
[3]NIHR CLAHRC for Northwest London, London, UK
[4]Department of Public Health and Primary Care, Imperial College London, London, UK

**Correspondence to**
Dr Laura Lennox;
l.lennox@imperial.ac.uk

## ABSTRACT

**Objectives** Although improvement initiatives show benefits to patient care, they often fail to sustain. Models and frameworks exist to address this challenge, but issues with design, clarity and usability have been barriers to use in healthcare settings. This work aimed to collaborate with stakeholders to develop a sustainability tool relevant to people in healthcare settings and practical for use in improvement initiatives.

**Design** Tool development was conducted in six stages. A scoping literature review, group discussions and a stakeholder engagement event explored literature findings and their resonance with stakeholders in healthcare settings. Interviews, small-scale trialling and piloting explored the design and tested the practicality of the tool in improvement initiatives.

**Setting** National Institute for Health Research Collaboration for Leadership in Applied Health Research and Care for Northwest London (CLAHRC NWL).

**Participants** CLAHRC NWL improvement initiative teams and staff.

**Results** The iterative design process and engagement of stakeholders informed the articulation of the sustainability factors identified from the literature and guided tool design for practical application. Key iterations of factors and tool design are discussed. From the development process, the Long Term Success Tool (LTST) has been designed. The Tool supports those implementing improvements to reflect on 12 sustainability factors to identify risks to increase chances of achieving sustainability over time. The Tool is designed to provide a platform for improvement teams to share their own views on sustainability as well as learn about the different views held within their team to prompt discussion and actions.

**Conclusion** The development of the LTST has reinforced the importance of working with stakeholders to design strategies which respond to their needs and preferences and can practically be implemented in real-world settings. Further research is required to study the use and effectiveness of the tool in practice and assess engagement with the method over time.

### Strengths and limitations of this study

► Feedback received from potential users of the Long Term Success Tool (LTST) throughout its development allowed us to design an approach that has responded to user preferences and addressed issues with language, length and practicality.
► The LTST builds on established literature and aligns well with other sustainability models but is distinguished from other approaches by its practical design and ability to draw on team suggestions for action planning.
► A systematic review of the literature may have strengthened our approach and uncovered further articles, but due to the practical time constraints of our programme, this was not possible.
► A limitation of this work is the potential for responder bias throughout development stages.
► Prior relationships between researchers and participants was identified as a possible source of bias, as participants may have responded in ways that were seen as more desirable to the researchers.

## INTRODUCTION AND OBJECTIVES

Significant financial and human resources are invested in initiatives to improve the quality of healthcare and deliver better patient outcomes. While many initiatives show patient benefits or improvements in care processes or clinical outcomes initially (eg, in the period when resource is available to introduce new practice), these often fail to sustain in the longer term.[1–5] As a result, there is growing research interest in this area, with studies showing wide variation in the sustainability of initiatives. Self-reported measures have shown that up to 60% of programme sustain (at least in part), while studies using more objective measures of sustainability

(such as independent observation) report lower rates of sustainability from 6.7% to 45%.[3 6]

This area of research is further complicated by several definitions of sustainability in the literature and little consensus on what constitutes 'achieving sustainability'.[1 7] Despite these issues, three domains of sustainability have been consistently used within the literature; continuation of initiative activities (maintenance of the intervention or practices that were introduced), continuation of the health benefits which resulted from the initiative (health outcomes remain stable or get better) and capacity built in the workforce (the skills gained by being involved in the initiative that can support ongoing high-quality care or the attainment of skills which enable the workforce to continually improve).[1] Given the complexity and dynamic nature of healthcare and healthcare delivery, we believe that all three domains are necessary to define and assess sustainability. For these reasons, we have chosen to define sustainability as: a dynamic process where staff and others involved have the capacity and capability to monitor and modify activities and interventions in relation to the health benefits they wish to achieve and in response to threats and opportunities that emerge over time. As sustainability is being seen as a process and not an end point, this definition does not include a specific time frame for sustainability. Time frames should be defined by initiative teams and stakeholders and be based on the goals of the improvement initiative with respect to the intervention, desired outcomes, disease area and setting.

Navigating the relationship between achieving initial 'successful' implementation and achieving long-term sustainability is a challenge.[1 8–10] It has been noted that over 60% of implementation frameworks include sustainability stages.[11] Factors contributing to sustainability of improvement initiatives often relate to how the improvement initiative is planned and conducted from the outset, suggesting an interdependent relationship between factors that influence initial success and those that influence long-term success.[1 8 9] Although the evidence shows an overlap in factors influencing both implementation success and sustainability, there is lack of clarity on what conditions may result in initial success but may or may not result in the sustainability of improvements. For example, an initiative may achieve initial success by providing extra resource or putting pressure on the workforce, but once the resource or pressure are removed the benefits achieved are not sustained.

## Addressing sustainability in practice

In the current healthcare climate of increasing demands and competing priorities for resources, healthcare planners and stakeholders are increasingly concerned with the long-term impact of their investments.[3 10] This has highlighted a need to understand how sustainability of improvement initiatives can be influenced and how specific approaches may help support sustainability.[3 10]

Defined procedures for addressing sustainability in improvement initiatives do not exist but many have suggested that sustainability indicators or factors can be used to monitor and influence sustainability over time.[1 4 12–14] Multiple strategies and approaches such as models and frameworks have been created to highlight such factors, but issues with tool design and content have been identified as barriers to their use in healthcare settings.[10 15–18] Specifically, poorly designed constructs, inadequate coverage of items and lack of clear definitions have impacted application and outcomes in past use.[15–18] Using methods well in practice is a recognised challenge for improvement teams, highlighting the need for all methods to be designed to be practical for use in real-world healthcare settings.[19–22]

The application of one sustainability method, the National Health Service Institute for Innovation and Improvement Sustainability Model (SM), has been previously described.[8 23] The SM is a self-assessment tool that details key factors that increase the likelihood of sustainability and continuous improvement.[24] The model is used to raise awareness of 10 factors for sustainability and prompt teams to consider actions to increase the likelihood of sustainability.[24] Application of this model demonstrated that while the SM raised awareness of determinants of sustainability and was perceived as valuable, teams found it difficult to understand and to apply the model routinely.[8 23] In particular, concerns were raised about the clarity the language used within the model, the user friendliness of design, the length of time taken to complete the questions and suitability for continuous use in healthcare settings.[8]

The purpose of this study was to collaborate with stakeholders to develop a sustainability tool relevant to people in healthcare settings and practical for use in improvement initiatives. To inform the tool development, we explored the following research questions:

1. How do sustainability factors identified in the literature resonate with the experience of those in improvement projects in healthcare?
2. What features or characteristics make a sustainability tool valuable, practical and useful in real-world healthcare practice?

## DESIGN
### Setting

Research was conducted within the National Institute for Health Research (NIHR) Collaboration for Leadership in Applied Health Research and Care for Northwest London (CLAHRC NWL).[25] CLAHRC NWL improvement projects cover a range of health problems and disease areas that include primary care, secondary care and community settings that are delivered over 18–24 months with the aim of sustaining improvements beyond this period. To support multidisciplinary teams to implement changes CLAHRC NWL systematically applies Quality Improvement (QI) methods such as the Model for Improvement and Action Effect Method.[19 23] The approach previously included use of the SM (2008–2013) but following internal

evaluation and published research, it was acknowledged that a new more user-friendly method for sustainability was required to meet the needs of improvement teams.[8 23]

## Participant information

Participants in this study included members of CLAHRC NWL improvement initiative teams and staff. These members come from various backgrounds: multidisciplinary healthcare practitioners (doctors, nurses, allied healthcare professionals), patients, carers, healthcare managers, directors, analysts and researchers (many participants hold overlapping roles, ie, nurse who is also a project manager). Other participants were also included at the engagement event and piloting. Although the majority of attendance is from improvement teams, these events were open to the public so additional participants included students, fellows, community members and industry partners. Specific participation from these groups is outlined within each development stage and summarised in the results.

## METHODS

Tool development was conducted in six stages. The first three stages: scoping review, group discussions and the stakeholder engagement event focused on reviewing the literature findings and their resonance with stakeholders in this setting. The last three stages: interviews, small scale trialling and piloting contributed to designing and testing usability of the tool. The researchers within this study had participant observer roles.[26] They provided teaching, facilitation and explanation throughout the development stages.

## Scoping literature review

A scoping literature review was undertaken to examine the extent, range and nature of research activity related to sustainability approaches.[27] The *r*esearch question guiding this review was: 'what approaches have been proposed to assess sustainability in healthcare and what sustainability factors are examined in each method'? *Identifying relevant studies:* A number of reviews had previously been published to identify factors for sustainability.[3 6 28] These reviews were used as a starting point to identify relevant authors and publications including snowballing of relevant journal articles, reference lists and the PubMed options of 'similar article' and 'cited by-' articles. *Selecting studies:* We sought approaches (published models, tools, strategies and frameworks) that identified sustainability factors and themes. Papers that introduced or described a sustainability approach were included. Papers only defining or constructing concepts of sustainability outside of a structured approach were excluded. Commentary, posters, protocols, conference proceedings, editorials and perspectives were excluded. *Charting the data:* A data extraction form was developed for identified articles. Data extraction included: approach name, approach purpose, year published, type (model, scale,

tool, checklist, framework), sustainability themes identified and scoring mechanism. One author (LL) screened the retrieved papers for inclusion and extracted the data from the articles. Data extraction was independently checked against the full-text articles by a second author (CD). Any discrepancies were discussed between authors and were resolved by consensus. Inclusion and exclusion criteria were refined to reflect these discussions Agreement was reached for accuracy of all studies. *Summarising the results:* All sustainability constructs (factors, questions, criteria, etc) identified in the approaches were extracted for thematic analysis. Aggregate themes were developed by combining similar or overlapping concepts and removing duplicate or redundant labels. Overarching sustainability themes were created using a mapping software.[29]

## Group discussions

Three facilitated group discussions were held with CLAHRC NWL team members to understand the perceived relevance of the literature review results against CLAHRC NWL team expertise and experience. Discussions were held during a weekly CLAHRC NWL meeting between core staff. The themes from the scoping review were provided on paper hand-outs to the attendees and an open discussion took place to determine the resonance and clarity of the themes presented. Observation notes were taken during group discussions. Notes were transcribed and findings were discussed among the research team to inform iterations of language and representation of themes which were iteratively adapted and presented at consecutive discussions.

## Stakeholder engagement event

Consolidated sustainability themes were presented to stakeholders at a CLAHRC NWL Collaborative Learning event in April 2014 to check the relevance and language against stakeholder views. In facilitated group discussions, participants provided their views on the resonance of these themes as well as identified any missing themes not seen in the literature. Designated note takers captured key learning and suggestions from the discussions. Field notes were collected and transcribed by one researcher. Findings were summarised and fed back to the research team to inform next steps and tool iteration.

## Interviews

Interviews aimed to collect in-depth information on value and practicality of tool design. A purposive sampling strategy was used to recruit interviewees. Participants were selected based on their role within diverse CLAHRC NWL improvement projects, their level of knowledge of their project and their experience with the SM (we sought both those with and without experience in using the SM to ensure we had a balanced sample of those with prior opinions of the SM). This approach aimed to maximise the diversity of perspectives gained from the interviews.[30] All interviews were carried out face-to-face in a workplace setting by one author (LL). A semistructured interview

guide was used for all interviews. The interview guide used open ended questions on tool value and features that would be most or least desirable to identify interviewee priorities. Interview questions explored the design of questions and statements used to draw attention to factors for sustainability as well as views on collating and presenting data to facilitate discussion and action. No specific questions on the sustainability themes were asked as the themes and factors had undergone two iterations with participant comments so further in-depth study was deemed unnecessary. The final interview question showed participants an early mock-up of the tool design on which they commented freely. Interviews were audio recorded and uploaded onto qualitative software Nvivo V.9. Audio recordings were coded directly on Nvivo using thematic content analysis.[31] A preliminary coding structure was developed using the interview questions as coding nodes, with themes inductively derived to summarise responses and record patterns in the data. The coding structure was iteratively developed, integrated and refined as further interviews were added to the dataset.[32] Results have been summarised using descriptive summaries and example quotes with explicit links to source text.

### Small-scale trialling
A group of individuals involved leading QI projects as part of a CLAHRC NWL fellowship programme were asked to trial a draft version of the tool. Trialling with this group aimed to understand the practical application of the tool including the approximate amount of time to complete by a wide range of people with diverse experience and expertise in improvement initiatives. Each participant filled out the tool for their own QI project. After completion, the group discussed the experience and posed questions on use. Critical feedback and suggestions for tool development were recorded as observation notes and summarised by the research team to inform tool iterations and piloting.

### Piloting
The resulting tool was piloted in July 2014. Piloting aimed to provide an opportunity for further comments and suggestions on practicality of the tool in healthcare settings, and to measure if the tool could be completed within an acceptable time frame. A brief presentation given to participants to outline tool design and instructions for use. Participants were asked to fill out the tool for their individual QI projects. Individuals without a formal project were asked to fill out the tool with a hypothetical project in mind. Participants were given 15 min to complete the tool and a 20 min facilitated group discussion followed. Designated note takers recorded key observations and feedback to inform tool iteration.

### RESULTS
Each development stage allowed for iterative adaptation and refinement of concepts, content and design of the tool. Key iterations from each stage are summarised in figure 1. The number and roles of participants is outlined in table 1. The following section discusses results from each development stage and concludes with an introduction to the resultant tool.

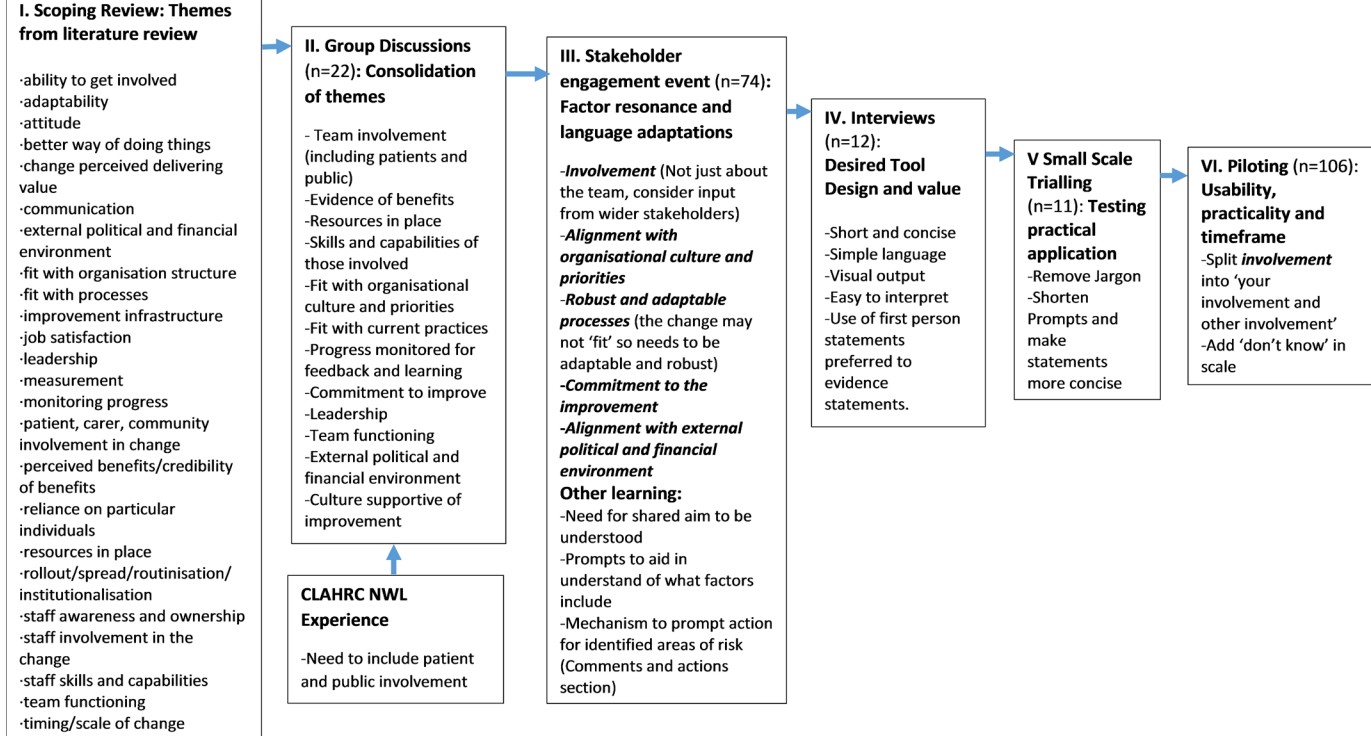

**Figure 1** Tool development stages and iterations

**Table 1** Number of participants by roles

| Development stage | Number of participants by role | | | | | | | |
|---|---|---|---|---|---|---|---|---|
| | Multidisciplinary healthcare practitioner | Healthcare or project manager | patient or carer | Researcher or academic | Student/ fellow | Data analyst | Other | Total |
| I. Scoping review | — | — | — | — | — | — | — | 0 |
| II. Group discussions | 5 | 9 | 0 | 3 | — | 3 | 2 | 22 |
| III. Stakeholder engagement event | 22 | 12 | 10 | 8 | 7 | 1 | 7 | 74 |
| IV. Interviews | 6 | 3 | 1 | — | — | — | 2 | 12 |
| V. Small-scale testing | 6 | 1 | 3 | 1 | 11 | — | — | 11 |
| VI. Piloting | 30 | 16 | 17 | 6 | 8 | — | 20 | 106 |

## Scoping review

The scoping review identified 81 publications for potential inclusion. Titles and abstracts were examined and 35 articles were retrieved in full text for full documentary analysis. Of these, 19 were excluded (16=no sustainability approach identified, 3=protocol, commentary or conference poster). In total, 16 publications which identified sustainability approaches were included in this review. The sustainability approaches consisted of six models, five frameworks, four tools and one scale. The approaches aimed to evaluate sustainability, plan for sustainability or provided guidance to study or influence sustainability of initiatives. Thematic analysis identified 25 overarching themes impacting sustainability (figure 1). Online supplementary appendix A summarises the approaches found and the sustainability constructs extracted. Results demonstrated reasonable consensus in the literature on factors influencing sustainability. The review uncovered themes not explicitly covered in the SM such as considering resources for the improvement, and the impact of the wider environment on initiatives. No strategy explicitly included the importance of involving patients or carers as an aspect of sustainability which was a key finding from previous CLAHRC NWL work.[8]

## Group discussions

In total, 22 individuals participated in the internal CLAHRC NWL group discussions. Discussions lead to combining themes that had different labels but were seen as having related or overlapping definitions. Discussions also identified where themes may be confusing and need to be expanded to underlying concepts to be relevant to improvement setting. For example, the literature theme of 'staff skills and capabilities' was expanded to include skills and capabilities of all those involved which may include as patients, carers or other stakeholders who participate in QI projects. Academic jargon and terms were also removed such as 'routinisation' which were seen as unhelpful or potentially confusing. These discussions resulted in changes to the language used and theme consolidation to form a list of 12 factors impacting sustainability (figure 1).

## Stakeholder engagement event

These factors were presented to stakeholders (n=74) in April 2014. The majority of the factors resonated well with stakeholders and were recognised as relevant to healthcare settings but in some cases the factor language needed to be adapted to align with stakeholder expertise and understanding. For example, the factor, 'Fit with Current Practice' was found to be problematic for participants. Although this factor was meant to convey the importance of interventions being aligned with current practice, many stakeholders mentioned that often improvements must be different from the current ways of working so trying to fit in with 'current practices' would not be desirable or possible. The factor was changed to 'robust and adaptable processes', highlighting the need for interventions with the ability to adapt to local settings.

Stakeholders also identified missing concepts and concepts they felt were not clearly represented in the current factors. For example, establishing a shared aim for a project was suggested as an explicit prompt underlying the factor 'commitment to the improvement'. Desirable design elements were also highlighted. Participants stated that team members entering scores should have the opportunity to comment and suggest actions to improve the prospects for sustainability. They suggested that comments could be brought together for each factor to provide a starting point for action planning based on team member ideas. Suggested changes were used to adapt language and definitions as well as inform design of the scoring mechanism of the tool.

## Interviews

Interviews (n=12) allowed detailed views from diverse stakeholders to be identified. Interviewees represented perspectives from multiple QI projects. Projects included frailty assessment in acute care, patient experience measurement for sickle cell disease, clinical pathway development for allergic conditions in children, medication review in the elderly, bundle development for chronic obstructive pulmonary disease and diabetes education in community settings.

Interviewees discussed sustainability measurement, tool value and functionality. Stakeholders unanimously expressed a desire for a tool that is simple to use and quick to complete:

> Brevity I think is the theme. It is very hard to have yet another form to fill or another algorithm to think about, for people who are already over worked and over stretched.(I3)

Interviewees desired a flexible tool with the option of quick review of the factors with any guidance or supportive text being brief and concise. Participants felt that using reflective statements to illicit an overall rating was a good way to get people thinking and provide an engaging format for the tool:

> I think overall impressions are powerful. You get a general feel and I think that is all you can hope for because otherwise…it will not be possible to make it user friendly. (I4)

The data and outputs used to stimulate discussions needed to be simple to access, interpret and present back to team members:

> I think most clinicians are familiar with a RAG (Red, Amber, Green) rating system so that would be easy for people to understand quickly. (I7)

From this feedback a draft tool was developed.

### Small-scale trailing

CLAHRC NWL fellows (n=11) trialled a draft version of the tool in June 2014. Each fellow was undertaking a QI project across diverse topic areas and settings (for example, service redesign, app development, patient experience measurement and staff training package development). Trialling the tool resulted in refinement the tool's prompt text to reduce the overall length. Stakeholders commented that the tool was a good reminder what to consider for sustainability but suggested changes to some of the language within the tool to remove terms perceived as 'jargon'. For example, in the factor 'Resources in place' original prompt text read: 'I am given sufficient headspace and time to dedicate to the improvement', after discussion the term 'headspace' was removed as it was seen as confusing to some participants. All participants completed the tool within 15 min. This time frame was discussed and seen as acceptable, with the fellows commenting that no more than 15 min should be allotted for routine tool use in practice.

### Piloting

The tool was piloted with 106 participants (83 of which returned a completed tool to the research team). Fifty-two participants indicated that were involved in active QI projects. This included 9 CLAHRC NWL QI projects across diverse topics (such as sickle cell disease, allergic conditions in children, polypharmacy in the elderly, chronic obstructive pulmonary disease and congestive heart failure) as well as 19 projects outside of the CLAHRC NWL programme. Piloting with stakeholders demonstrated that majority of participants completed the tool in the projected 10–15 min time period. Stakeholders engaged well with the prompts within the tool, commenting that they provided a simple format to begin consideration on how each factor may impact their initiatives. Participants commented that the tool was easy to use and that the statements and questions enabled good discussion and 'promoted deeper thinking' allowing them to think about things they had not previously considered.

Regular scoring and review of factors was discussed and participants agreed in the necessity of consistently reviewing the changes to sustainability throughout their initiatives. Use every 3 months was recommended by stakeholders, as they felt this time frame would be feasible given the ease and design of the tool and the potential for changes and turnover of staff in settings. Participants suggested the addition of a 'don't know' and 'no opinion' option to the tool as they did not want to make a forced choice and rate a factor that they did not have enough information to make an accurate rating. During piloting, stakeholders questioned the appropriateness of the term 'sustainability'. Many stakeholders felt that 'sustainability' did not accurately capture the need for potential adaptation of initiatives or the desire to continually improve practice. Stakeholders wanted a term that would include both sustained improvements as well as the long-term commitment to improvement. These discussions resulted in the term 'long-term success' being used in place of sustainability to represent the aim that stakeholders desired. Feedback was used to iteratively develop the tool, which was then rolled out for wider use by CLAHRC NWL teams in January 2015. The final design of the tool and description for use is discussed below.

### THE LONG TERM SUCCESS TOOL (LTST)
### Purpose

The LTST aims to support those implementing improvements reflect on 12 key factors to identify risks and prompt actions to increase chances of sustainability over time.

### The factors

The factors included in the tool are: commitment to the improvement, involvement, skills and capabilities, leadership, team functioning, resources in place, evidence of benefits, progress monitored for feedback and learning, robust and adaptable processes, alignment with organisational culture and priorities, support for improvement and alignment with external political and financial environment. The factors and their effects have been well documented in the literature.[1 3 6 33] The presentation and language used to articulate the factors has been carefully developed and adapted with stakeholders to improve ease of understanding and user friendliness. The 12 factors have been organised within 3 emergent areas; people, practice and setting. Table 2 describes the factors and

**Table 2**  Long-term success factors: purpose, statement for rating and questions to consider

| Factor | Purpose | Statement | Additional questions to support reflection |
|---|---|---|---|
| **People** | | | |
| 1. Commitment to the Improvement | To reflect on both own personal commitment to the initiative and impression of commitment across the team as a whole to the initiative | My team understands what the project is trying to achieve and believe this work will lead to improved processes and outcomes. | Do you feel committed to the project? Do you understand what the project is trying to achieve? Do you believe that this work will improve processes and outcomes? Do you believe there is reliable evidence (eg, from literature, guidelines, etc) that the project will produce the desired benefits? Do you think there is commitment across the team as a whole? Has a shared aim been established for your project? If you think commitment is lacking, what do you think is the reason for this? What do you think should be done to address this? |
| 2. Involvement | Reflect on who has been involved and who may need to be engaged further for the initiative to achieve long-term success. Asks about personal involvement and contribution and explores the involvement of patients, carers and members of the public who are impacted by the changes being made | I have the opportunity to input into the project and I feel a sense of ownership towards the work. I am able to express my ideas freely which are openly considered by the team. There is wide breadth of involvement from stakeholders including patients and members of the public who regularly feed into the project. t | Do you personally feel involved in the project? Are you given the opportunity to express your ideas and recommend changes to the project when necessary? Do you think the project has involved the right people? Does your project involve patients affected by the improvement? Is there involvement from staff who will be delivering the improvement as part of their day-to day practice? Are the views of these groups taken on board? Does the project have a good spread of views, skills and expertise? Do you believe involvement can be improved? Are there groups of people you still need to involve? |
| 3. Skills and capabilities | Explores whether the staff and other people delivering the change have the skills to do so successfully and whether training of new members of the team has been planned for | Staff have the necessary skills to deliver the improvement. Training and development opportunities are available to all staff, volunteers and other people involved. | Do you feel able to fulfil your role within the project? Do you require further training or education? Do staff who will be delivering the improvement (eg, front-line or support staff) have the skills to do so consistently and effectively? Are new staff informed about the project and their role in it? Do you think there are training needs associated with the improvement that should be addressed? What should be done to address to these needs? |
| 4. Leadership | Asks if there is strong leadership in place and if the leaders are approachable, available and able to garner support for the initiative | My project has supportive and respected leaders and/or champions who advocate for the improvement, communicate the vision and effectively manage the process. | Do you believe your project has strong leadership? Are your project leaders actively involved in the project? Are they able to garner support and enthusiasm for the work? Are they available and approachable to members of the team if necessary? Do the project leads effectively communicate the need for the change? How do you think leadership could be strengthened? |

**Table 2** Continued

| Factor | Purpose | Statement | Additional questions to support reflection |
|---|---|---|---|
| 5. Team functioning | Explores the accountability and responsibilities for the workload involved in the initiative and ask if the team is working well together | My project team is working well together. There are clear responsibilities for individuals and the work is shared across the team and does not rely on particular individuals. | How well do you feel your project team is working together? Does the project team meet and communicate on a regular basis? Have clear roles and responsibilities for project team members been established? In your opinion, are team members fulfilling these roles and responsibilities? Are skills and expertise of team members considered and put to use? What do you think can be done to improve team functioning? |
| **Practice** | | | |
| 6. Resources in place | Explores if the necessary resources such as staff time, equipment and facilities have been dedicated to the initiative | My project has financial support that will allow the improvement to achieve long-term success. We have the necessary staff, material and equipment and I am given sufficient time to dedicate to the improvement. | In your opinion, have enough resources been dedicated to support the project? Do you believe the financial support provided will allow the improvement the project is trying to achieve to become part of normal working practice in the long term? Does the project have enough staff to achieve the project aims? Do staff have enough time to spend on the improvement? Are the materials needed (eg, physical facilities, sites, equipment, etc) available to staff when they need them? Are resources needed discussed by the team on a regular basis? What resources do you think are lacking? |
| 7. Progress monitored for feedback and learning | Encourages teams to consider what systems are in place to monitor the initiative over time and how this information will be used to inform staff of further changes needed | There is a monitoring system in place that allows the team to collect, manage and regularly review data. Feedback from the project is shared with me and other stakeholders on a regular basis. | Have measures to enable continuous monitoring for your project been defined by the team? Do you think these measures are able to assess the impact of the improvement? Can you suggest any changes to improve this? Are these measures regularly assessed? Is this information used to make changes and improve project progress? If the measures show lack of progress are the causes for this investigated? Are project members and staff regularly informed about what is working well and what could be better? |
| 8. Evidence of benefits | Asks if and how the benefits of the initiative are communicated to both staff and patients over time | There is evidence of benefits emerging from the project and this evidence is regularly communicated and visible to staff and patients. | Does the evidence for your project include both the impact on physical and mental well-being of patients? Is there evidence (process and outcome measures) that the project is producing the desired impact on patients? Is evidence of the projects' impact regularly shared with staff, patients and other stakeholders? If evidence shows lack of progress, does the team investigate reasons for this? |

Continued

**Table 2** Continued

| Factor | Purpose | Statement | Additional questions to support reflection |
|---|---|---|---|
| 9. Robust and adaptable processes | Reflects on the need for initiatives to be adapted to local processes and emerging needs. It also asks about the process for recording successes and failures of changes made | There is the opportunity to adapt the project to reflect local needs, setting and emerging evidence. Adaptations are documented and the successes and failures of changes are reported. | Is there regular review of how the project is working? How well does the project fit within current practices? Do staff and team members need to adapt how they implement the improvement in response to challenges or changing care needs? Does your team use Plan Do Study Act cycles, Statistical Process Control and other quality improvement methods to test and document the changes made to the improvement? |
| **Setting** | | | |
| 10. Alignment with organisational culture and priorities | Encourages teams to consider the need to align improvement initiatives to organisational strategies to gain executive buy-in and support as well as have the initiative become part of organisational policies and procedures | The improvement my project is trying to achieve is aligned with the strategic aims and priorities of the organisation(s) we work within and our work contributes to these aims. Our work is supported by the policies and procedures within the organisation. | Is the improvement your project trying to achieve aligned with the organisational priorities? Has this been promoted as something to help further the organisation's aims and priorities? From your perspective, how well is the work of the project being integrated into the everyday operations of the organisation? Does the project conflict with any other changes taking place within the organisation? What could be done to better align your improvement to these priorities? |
| 11. Support for improvement | Explores the values and beliefs held within organisations related to continuous improvement and looks at the support given to staff and patients to be involved | There are values and beliefs in my organisation(s) that emphasise the need to improve. Staff and management are supportive of improvement initiatives and continuous improvement is a priority for the organisation, staff and patients. | Do you feel continuous improvement is a priority within your organisation? Are staff and senior management receptive to improvement initiatives? Are you supported by your leaders to participate in the improvement initiatives? Do senior leaders actively participate in improvement projects? |
| 12. Alignment with external political and financial environment | Looks at the need for teams to be aware of the potential political and financial changes that may impact the initiative | My project exists in a supportive economic and political environment. My team is aware of external pressures and incentives that may influence the project. | Has your team considered the impact of the external environment on the project? For example, are there economic pressures or political developments that may impact the project? Is there political support for the implementation of your project? Does your project help address external political or economic concerns? Does it contribute to the achievement of political objectives? Are there plans to mitigate risks due to the external environment? |

provides the statements for rating and supporting questions included within the tool.

## How it works

The LTST is designed to create a platform for people to share their own views on sustainability as well as learn about the different views held within their team and to prompt discussion on any difference in opinion. To ensure teams are aware of how systems are evolving over time, teams are encouraged to use the tool approximately every 3 months to assess progress and identify emerging risks continuously. Team members are asked to provide their overall impression of how their team is performing in each factor. Responses are collected on a paper questionnaire form or on the CLAHRC NWL Web Improvement System for Healthcare.[34] The full paper questionnaire can be found in online supplementary appendix B .

For each factor, team members are provided with a statement intended to prompt reflection. Supporting questions are available for each factor if team members would like more detail on what to potentially consider (table 2). Team members score each factor individually and anonymously using a simple five-point Likert scale (as well as no opinion and do not know options). Team members provide comments to suggest actions, explanations of their rating or concerns about progress against each factor.

Team scores are then brought together to produce aggregated outputs demonstrating how the initiative is performing against the given factors. Figure 2 shows an example a visual chart produced highlighting risks and differences in opinions. Table 3 shows an example of aggregated comments and actions highlighted within the tool.

Visual charts and comments are intended to facilitate discussion, bring differences of opinion or concerns into the open and encourage actions to increase the chances of improvements being sustained. For CLAHRC NWL teams, time is allocated at progress meetings to review scores and plan actions.

## DISCUSSION

The aim of this work was to develop a relevant and practical tool for sustainability that meets the needs of people in improvement initiatives. We explored how sustainability factors identified in the literature resonated with those in improvement projects and the features or characteristics which make a sustainability tool most valuable in real-world healthcare practice. This work has shown that the majority of factors from the literature resonated well with stakeholders and were recognised as relevant to healthcare settings. In some cases, the literature findings needed to be adapted through changes to the language used to align with stakeholder preferences and understanding. Engaging stakeholder in the design process demonstrated that stakeholders valued clarity, conciseness and simplicity for tool design with simple data interpretation and visual graphs. Receiving ongoing feedback during the development period from those who will use the tool has allowed us to design an approach that has responded to user needs and has addressed issues with language, length and practicality along the way.

The LTST provides a mechanism for improvement teams to identify risks to sustainability and importantly can create an environment for team members to highlight specific actions to be taken and comment on ways to influence sustainability over time. The LTST builds on established literature and aligns well with other sustainability models and frameworks with all LTST factors reflected in one or more of the other approaches.[1 2 4 9 24 35–44] LTST is distinguished from other approaches by its practical design and ability to draw on team suggestions for action planning. Using participant ideas as a platform for action is a unique feature of the tool that is not present in other tools currently used in this area. Also unique to the LTST

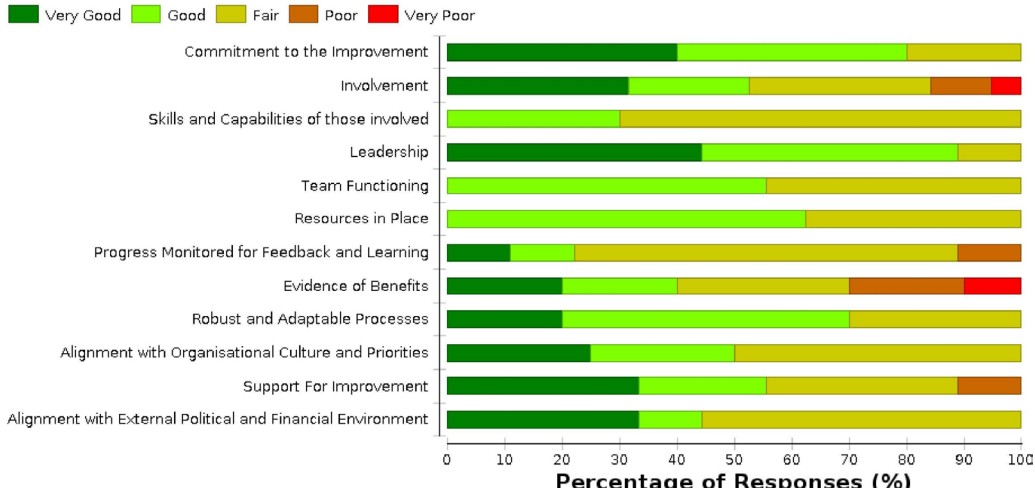

**Figure 2** Team level graph highlighting areas where the team is doing well, where more work is needed and differences of opinion.

**Table 3** Comments and actions provided by team members during scoring which can be a starting point for discussion

| Factor | Comments and actions |
|---|---|
| Commitment to the improvement | ► Clear summary of project components and effects now in place from last time<br>► Make sure all stakeholders attend meeting<br>► As a commissioner I did not understand expectations and my role in the group—others seem very committed.<br>► Need to look at those engaging with the project |
| Involvement | |
| (a) | ► Difficulties moving forward as until all stakeholders are engaged—unable to move forward<br>► Need to consider who is not involved and who would bring influence and value to the project |
| (b) | ► More patient/parent engagement at local level helpful<br>► More needed<br>► Patient and public involvement needs to be broadened<br>► No public/patient—do not feel it would be appropriate<br>► Patients/patient group and primary care practices poorly represented |
| Skills and capabilities of those involved | ► Of current clinical staff that I am aware of<br>► Capacity issues potentially can limit progress<br>► More nurse input<br>► Not enough nursing staff employed to deliver project currently<br>► Needs consultant/general practitioner and nurse shadowing and specific training<br>► Limited number of staff needs expansion |

is that the allotted time for use, an identified barrier and challenge to other method use, has been explicitly tested and informed by end users.[8 45] While many other methods involve either unknown or substantial time commitments, the LTST can be completed in approximately 10–15 min.[42 45]

There is also potential to supplement the use of other models or frameworks to complement the LTST. For example, if a project receives a low rating for the factor 'robust and adaptable processes', the Model for Highly Adoptable Improvement tool kit may be used to aid the team in further understanding of where the intervention can be adapted.[46]

## LIMITATIONS

A limitation of this work is the use of a snowballing scoping review opposed to a systematic review. Conducting a full systematic review may have uncovered further articles and/or approaches, but due to the practical time constraints of our programme, this was not possible. The results of our review have fed into a protocol for a full systematic review on available sustainability approaches which is now under way.[47] The results of this review will inform future adaptation of the LTST.

Another limitation of this work is the potential for responder bias throughout development stages. Prior relationships between researchers and participants was identified as a possible source of bias, namely, social desirability bias, as participants may have responded in ways that were seen as more desirable to the researchers.[48] Another source of possible responder bias stems from the sustainability themes and factors being presented to participants during development stages which may have directed participant responses and reaction. Although participants were given the opportunity to provide their views on the resonance of these themes as well as identify alternative themes, participants may have been more likely to agree with presented findings which may have impacted our findings. As the development of the tool was centred on user preferences, attempts were made to communicate and reiterate there were no 'right' answer to questions. We also attempted to mitigate this effect by having multiple stages for feedback, with diverse facilitators and a wide variety of participants. We also had a researcher unknown to the majority of the interviewees conduct the interviews.

Another possible limitation is related to the generalisability of the tool to teams with little or no QI experience. Although the tool was developed by people with significant QI experience, the tool is intended to be used by those with all levels of QI knowledge. The process of involving those would use the tool in the design and piloting of the tool sought to ensure the tool could be used by all people involved in an improvement project. Tailoring of the tool language and the instructions were done to ensure people with little QI experience or knowledge would be able to use the tool. Further observation and study of the application of the tool is needed to assess if application is impacted by this design.

## FUTURE RESEARCH

While attempts have been made to respond to user preferences and create a practical tool, further research is required to assess tool effectiveness and engagement

## Box 1    Information on using the Long Term Success Tool

### Using the Long Term Success Tool in your setting

► The Long Term Success Tool (LTST) has been designed on the Collaboration for Leadership in Applied Health Research and Care for Northwest London Web Improvement System for Healthcare system. For those who do not have access to this system, the LTST questionnaire form and Excel spreadsheet can be downloaded with this paper. The tool can be used along with table 2 which provides supporting questions to describe the potential items to consider within each factor. The tool can be used by individuals and teams. Responses can be input into the Excel spreadsheet which enables users to produce similar graphs and outputs to ones shown in this paper (supplementary appendix C). The spreadsheet enables eight possible entry points for a team (up to 20 team members) and will aggregate team data over time for review and action planning (see online supplementary appendices B and C).

over time. A 3-year programme of research with teams at CLAHRC NWL and other groups internationally is currently under way to investigate tool impact on initiative processes and practices and examine actions taken by improvement teams to sustain improvements across diverse settings and environments. This longitudinal study will also investigate tool links to sustainability outcomes to assess what impact tool use may have on sustained QI projects. To facilitate and study the use of the tool by those outside of Northwest London, the tool is freely available along with a structured excel spreadsheet for data entry to produce automated graphs and charts (box 1).

### CONCLUSION

The development of the LTST has reinforced the importance of working with stakeholders to design strategies which respond to their needs and preferences and can practicality be implemented in real-world settings. This study provides valuable information on the process of developing a new approach to sustainability that is both conceptually rigorous and practical for use with healthcare improvement teams .

### ACKNOWLEDGEMENTS

The authors would like to thank Dr Alan Poots for his contribution to designing and developing the LTST spreadsheets. We would also like to thank the stakeholders involved throughout the development process.

**Contributors** All authors were involved in conceiving the study. LL conducted the interviews. All authors contributed to data collection in the group discussions and piloting. CD, LL and JR conducted the analysis. LL drafted the initial manuscript. All authors contributing to revisions and the final draft of the manuscript.

**Funding** This work was funded by the National Institute for Health Research in the Collaboration for Leadership in Applied Health Research and Care for Northwest London (CLAHRC) programme.

**Disclaimer** This article presents independent research commissioned by the National Institute for Health Research (NIHR) under the Collaborations for Leadership in Applied Health Research and Care (CLAHRC) programme for North

West London. The views expressed in this publication are those of the author(s) and not necessarily those of the NHS, the NIHR or the Department of Health.

**Competing interests** None declared.

**Ethics approval** Ethics approval was not required for this work as it was part of a service evaluation and improvement project. All interviewees provided verbal consent for the recording of the interviews and were informed that all data would be anonymised for publication.

**Provenance and peer review** Not commissioned; externally peer reviewed.

**Data sharing statement** No further data are available.

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
