## [Reviewer comments · BMJ Open]

ARTICLE DETAILS

TITLE (PROVISIONAL)	What makes a sustainability tool valuable, practical, and useful in real world healthcare practice? A mixed methods study on the development of the Long Term Success Tool in Northwest London
AUTHORS	Lennox, Laura; Doyle, Cathal; Reed, Julie; Bell, Derek

VERSION 1 - REVIEW

REVIEWER	Samuel Silver University of Toronto, Canada
REVIEW RETURNED	18-Jan-2017

GENERAL COMMENTS	This study combined scoping review and qualitative methods to develop a tool to measure the sustainability of quality improvement work. I have a few questions and comments to clarify the methodology and usability of the tool: Major Comments: 1. Please provide more information on the pilot testing?• Was training provided to use the tool?• Was the tool piloted on actual QI projects or fictional QI projects?These questions relate to the generalizability of the tool, which should be discussed.2. The manuscript requires a discussion of study limitations. Some issues I noticed include:• The tool was developed by people with significant QI experience. Will this tool be generalizable to teams with less QI experience?• The study did not use Delphi methodology and comments were not anonymous, so how can the authors be certain all opinions were represented (especially since qualitative interviews did not reach saturation and many participants may have had a prior relationship with the interviewer and so restricted their comments)?• The tool assumes all sustainability factors are created equal, which is not always the case. Was an attempt made to determine which sustainability factors were most important?3. The authors should be more cautious on some of their interpretations, since the real-world efficacy of this tool still needs to be demonstrated. For example, it has yet to be proven that with actual QI projects among less experienced teams the tool is:• Quick to complete• Involves simple interpretation• Leads to suggestions to address sustainability issuesThe authors indicate this is the case on page 18, but I think more validation is needed to support this statement. At the very least, it
--

	should be stated that this was the experience in the pilot phase, and whether this will extend to actual QI is unknown. 4. Future plans for the tool should be discussed. Do the authors intended to validate it prospectively and in different environments? Will the authors evaluate whether use of the tool actually leads to more sustained QI? Minor Comments: 1. Abstract: The details of the scoping literature review methodology should be provided. 2. The “design” part of the abstract is not clear. I cannot tell the order in which feedback was elicited. Maybe using numbers would help. 3. Consider mention the roles of all stakeholders in the abstract (e.g., patients, carers, industry, government, etc.) 4. Page 8: Please list the number of the different stakeholder groups involved (e.g., number of patients and carers) 5. Page 11: Text is quite jumbled. Maybe consider presenting the different factors with bullet points or numbers? 6. Page 17, line 24: I think the authors mean to refer to Table 3? 7. Figure 3 does not add much. Please consider removing.
--	---

REVIEWER	MonDep. of Learning, Informatics, Management and Ethics, Karolinska Institutet, Sweden ica Nyström
REVIEW RETURNED	13-Feb-2017

GENERAL COMMENTS	It has been interesting to read the paper and get some insight into how you tried to develop a practical tool that is easy enough to use repeatedly b practitioners. However, I have had some difficulties grasping the mauscript, especially as a reserach study, and have tried m best to provide constructive suggestions. I hope this will aid further development of the manuscript! 1. Is the research question or study objective clearly defined? My main concern is the objective of the study that makes it difficult to separate it from a description of a development process. In the Abstract the objective of the study is described as: “This work aimed to collaborate with stakeholders to develop an evidence-based sustainability tool relevant to people in healthcare settings, and practical for use in improvement initiatives. On page 5 the purpose of the study is described as: “The purpose of this study was to develop a tool for improvement teams to plan, and reflect on, factors important to sustainability at all stages of an initiative, to prompt discussion and action to enhance chances of achieving sustainability.” For me this aim/objective/purpose is that of a development project rather than of a scientific study and there need to be something to study that we can learn from and a clear aim/objective/purpose for the study of the development at hand. Some not so elaborated examples that I can think of might clarify what I mean: 1. What factors contributed to x and x during the collaborative design process and how did it change the model? = focus of the study is to
--

provide knowledge on a collaborative design process and what it is that is important during such processes (co-design)

2. What are the differences/similarities between the model extracted from the literature review and the subjective experiences/views of professionals is another option (though the themes might have been studied by others already) = focus of the study is to compare the theoretically derived constructs/themes with users' understanding/experiences of what it is that is important for sustainability of improvement

3. The description of the literature review and how the first model/themes were derived might also act as a background to how the instrument was developed and then the focus is on a pilot study of user's perceived relevance and understanding of the derived concepts (themes) and perceived usability among their teams (and further studies, as suggested, will be on it's actual use and effect in empirical settings). Then the changes to the initial model will be described etc. (se under Study design below)

At page 5 you state that 'This article describes work undertaken with healthcare professionals, patients, and researchers to develop a tool to meet the needs of people in improvement initiatives'. Were all these stakeholders involved during the entire process or in specific parts (test of themes, test of form and usability)? In what way were patients involved?

2. Is the abstract accurate, balanced and complete?
It will need revision of objective (se above) – and based on these changes also under Results and Conclusions.

3. Is the study design appropriate to answer the research question?
The study design is somewhat confusing (also due the discussion on objective/purpose above) – it can be clarified by describing it as three steps/phases that also will clarify the Results section (both sections described below):

1. Literature review and consolidation into a model/theoretically derived themes on sustainability – Methods used (a more specific description of how themes were chosen)
Results = Model 1 themes (content)
2. Design process (interactive and iterative) – test of subjective relevance of Model 1 – Methods used, procedure over time, respondents (why this sample of potential users)
Results = Description of discrepancies, changes/adjustments made, and Model 2 themes (content)
3. Pilot test of Model 2 – both content and form
Methods used, procedure, respondents (why this sample of potential users)
Results = Description of perceived relevance and usability, changes/adjustments made, and Model 3 themes (content)

Also consider and discuss what you mean by developing and evidencebased instrument - it might not be an appropriate term to use depending on the influence of the process on the limited themes left in the final tool...

4. Are the methods described sufficiently to allow the study to be repeated?
The review started by going through 3 previous reviews (from 2005 and 2012). Criteria for including/excluding the other sources of information (the results of literature review, Appendix A) is less clearly described and some considerations that may inform further work on the paper and providing some argumentation behind the 12

factors included (and why other factors were excluded) are suggested below.

Data collection and informants

A clarification of the samples of informants/respondents is needed and why these were selected in each step – it is difficult to understand. Below some questions that popped up while reading the text. I think you need to provide some text on the participants in the CLAHRC NWL Improvement projects and their role/profession etc., maybe under Setting? Maybe they are both clinical staff and trained to participate in CLAHRC NWL Improvement projects?

Clarify who you refer to as 'stake holders' in this case and which method that is used to capture each type of stakeholder's experience/view. "CLAHRC NWL improvement initiative teams and staff, team members" – clarification regarding what/who you refer to would facilitate the reading experience (Do these improvement initiative teams not include staff and if so which staff? Clarify which staff are you referring to? Clinical staff? Staff at the CLARHRC NWL? Is this the same?)

Are the informants overlapping - participants in the 1) Facilitated group discussion, 2) the Collaborative Learning events, 3) interviews,) the Small scale trialing and 5) Piloting? Clarify... Maybe use a Figure/Table over all the Phases (se above) and each of the steps where data was collected (six steps with the scoping literature review). Why so few clinical staff (if this is the case) – especially when testing the perceived usability?

Why are you reporting sample size: (n=12) in table 1 (COREQ) but a larger number of participants in group-discussions and at the stakeholder event in the Methods section?

The role of the authors/researchers during data collection – active or just observing the discussions? Who led the group discussions, engagement events and small scale trialling?

Sampling of informants/respondents: Regarding the interviews you state that 'Participants were selected based on their role within the improvement project, their level of knowledge of the project, and their experience with the SM.' Clarify "improvement projects" (=CLAHRC NWL Improvement projects?) as this can be confusing for a reader. Maybe also elaborate just a little on why their experience of the SM is relevant?

Elaborate a little more on the information regarding the interview guide. No questions on content/themes and their relevance at this stage?

In the methods section you describe a 'Small scale trialing' (n=15) and a 'Pilot' (n=83) of the LTST. Where and with whom were these "tests" conducted? (In ongoing improvement efforts at work place settings? With clinical staff? Managers? Stakeholders? Process supporters?) The information provided regarding these two events leaves the reader with several questions.

Data analysis

You state (in table 1 COREQ) that your themes were derived from data. What is the difference between an identified theme and the listed factor from the literature? Are grouped factors = themes? A section describing the data analysis procedure is missing for all types of data collections (except maybe for the literature review) and information in table 1 does not describe all procedures.

A clearer description of if and how, the model/themes was re-designed during the development process – i.e. the emergence of new themes and the adjustments of all themes during the process so it is easy to follow the progress of the model/themes – and the

arguments/discrepancies the occurred during each phase/step. Figure 1 provides some of this information but this can be clearer. How were the observations analysed? When needed how were the data sources combined during the analyses? Did the design of group discussions allows for new themes to emerge or were the themes presented and then reacted to – if so what are the risks? = Discuss study and methodological limitations under the Discussion section.

5. Are research ethics (e.g. participant consent, ethics approval) addressed appropriately?

Ethical considerations are addressed according to the rules in the UK (?) I guess, maybe clarify the role of the researchers during data collection here or in the Methods section. This article type is defined as a research study and but I still think that ethical approval is not needed.

6. Are the outcomes clearly defined?

In the results section (page 11-12) you state that the 12 factors have been organized within 3 areas; People, Practice and Setting, and provide table 3 which describes the factors and the statements for rating and supporting questions included within the tool. Why (and for whom) were the factors organized in these three areas and how and when is table 3 used (e.g. is it part of your analysis or development process or is it for 'practical' use?).

7. If statistics are used are they appropriate and described fully?

Not relevant

8. Are the references up-to-date and appropriate?

Depending on changes in purpose/objective additional literature (co-design?) may need to be added.

9. Do the results address the research question or objective?

They describe the development process and the resulting tool – otherwise see comments on objective/purpose.

10. Are they presented clearly?

Depends on the more specific objective/purpose. I suggest a presentation in relation to Phases: 1) Literature review and consolidation into a model/theoretically derived themes on sustainability; 2) Design process (interactive and iterative); and 3) Pilot test of Model 2 – both content and form where the LSTS (Model 3) can be presented (and intended use of the LSTS tool (Model 3) is discussed in The Discussion section along with the changes during the design and piloting process and potential benefits and shortcomings this might have added in relation to the first model).

11. Are the discussion and conclusions justified by the results

Discussion

I also miss a reflection on how the resulting tool is similar/different from earlier theories/models and tools. For example, CFIR has toolkits that perhaps could be used in practical settings. Also, how much did the tool develop from the scoping review to the final tool? What came out from the different steps – and a discussion of the steps would help.

A lot of valuable information is provided in this section regarding your thoughts on when and by whom the LTST is supposed to be used. As a reader I would prefer an short introduction to your thoughts on this matter earlier in the manuscript, that is elaborate a

	bit more on the difficulties using the tools available and what they might be lacking. Methodological considerations/study limitations are not addressed adequately in this section. Conclusion section Needs to follow a clarified objective/purpose of the study. 12. Are the study limitations discussed adequately? Study limitations have not been addressed – this needs to be added.
--	---

REVIEWER	Diane Lorenzetti University of Calgary, Canada
REVIEW RETURNED	29-Mar-2017

GENERAL COMMENTS	Comments to Authors This paper describes the development of a stakeholder-driven tool to assess the ongoing and long-term sustainability of healthcare improvement projects and programs. As the sustainability of new initiatives is often a barrier to both implementation and ongoing support, a tool to aid decision makers and other stakeholders to conduct formative assessments of such initiatives is an important timely contribution to both practice, and the existing literature on this subject. BACKGROUND I appreciate the authors clearly articulating the definition of sustainability that guided their project and tool development. On page 5 the authors refer to a sustainability model developed by the NHS Institute for innovation and improvement sustainability. While the authors state that this model has been reported elsewhere, I would suggest it is important to include a brief description of this model here as well, particularly as it appears that this work may have at least partially inspired the present study. While I realize that the scoping review the authors have undertaken was meant to identify existing research on program sustainability, the inclusion of a few additional studies in the background section of this paper that speak to existing gaps in the literature would strengthen the rationale for this work. METHODS Scoping Review The authors have not included a detailed methodology for their scoping review. The Arksey and O'Malley framework adopted for this review outlines specific steps for conducting a comprehensive and transparent review. This framework includes: "identifying the research question, searching for relevant studies, selecting studies,
--

charting the data, and collating, summarizing and reporting the results, and consulting with stakeholders to inform or validate study findings". While no commonly agreed guidelines exist for the reporting of scoping reviews, authors often incorporate a level of detail in their reporting that is similar to that outlined in the PRISMA Guidelines for the Reporting of Systematic Reviews (<http://prisma-statement.org/>). This includes: eligibility criteria, a list of information sources, a detailed search strategy (or list of search terms used) and any search limits such as language or date, study selection process (which often incorporates dual review), and the charting process/thematic analysis that was adopted (including a partial list of data elements that were extracted). While some of these elements are present in this manuscript, I would recommend the authors include additional details on their approach.

In their 2005 article, Arksey and O'Malley stress the need to be "as comprehensive as possible in identifying primary studies (published and unpublished) and reviews suitable for answering the central research question". While consulting prior reviews, searching reference lists, and citation searching/related articles in PubMed are all useful approaches to study identification, comprehensive searching of relevant electronic databases is essential to conducting a scoping review. I can't tell if the authors constructed detailed search strategies to search PubMed, or relied on reference lists and snowball sampling of studies included in prior reviews. A detailed search of PubMed and other databases (such as EMBASE or CINAHL) might have yielded important frameworks and models that could have informed tool development. While I am not suggesting the authors expand their search, I do feel that it is important to provide additional details on the search process and address any search limitations in the Discussion section of this manuscript.

Group Discussions, Stakeholder engagement and Interviews.

Could the authors comment, in their manuscript, on prior relationships, if any, researchers had with focus group and interview participants?

The authors state that they adopted a thematic content analysis approach (page 7 line 42) to derive themes from their interview data. Can the authors please outline this process in more detail?

Trials: Small Scale and Pilot

Who were the individuals who participated in the small scale trial and pilot of the tool? Were they program developers, project coordinators, or others with relevant experience? Can the authors please provide additional detail on how these trials and pilots were conducted? I'm left wondering if participants were all scoring a standardized exemplar project/program or different ones, and how

	the type of program/project assessed may have influenced/or not influenced the feedback received on the utility of this tool? RESULTS The authors should include a statement outlining the number of studies identified and number of studies included in their scoping review. The authors employed a variety of steps and inputs to develop their tool. I appreciate the inclusion of a figure summarizing this process, and the changes that occurred to the tool throughout. On page 9 line 8 the authors state that “these discussions resulted in a consolidated list of 12 factors impacting sustainability”. Suggest citing Figure 1 here. Did these 12 factors change as a result of the stakeholder engagement event and interviews? A statement either way would be helpful. DISCUSSION The discussion should begin with a broad statement outlining what the authors accomplished in their study and any unique findings that were noted during the process of tool development. The tool the authors have developed could potentially also inform program/project planning efforts. The authors may wish to comment on this in their discussion. Do the authors themselves have specific plans to test their tool in practice and over time? OTHER On page 24, the header for Figure 1 is mislabeled as Figure 2. Table 1 consolidated criteria: In the Domain 2 (study design) section of this table, the authors should report on sample sizes and data collection methods for all stages of their study – not just interviews.
--	--

REVIEWER	Micah D J Peters Faculty of Health and Medical Sciences, The University of Adelaide, Australia
REVIEW RETURNED	05-Apr-2017

GENERAL COMMENTS	Introduction and objectives: - The introduction and objectives section clearly sets up the problem
---

	describing how there are significant challenges to the sustainability of healthcare improvement projects. Some limitations around researching sustainability are provided, for example issues with consistency in terms of key definitions and reporting. Key domains of sustainability are briefly introduced. An operating definition of sustainability is provided. Frameworks/models of implementation are introduced and some of the challenged teams face in using them in practice are briefly presented. The objectives of the project/paper are clearly explained and the parameters around the objectives provided.  - Minor suggestion: replace “patients” with “healthcare consumers” Design: Scoping Literature Review:  - A scoping literature review was undertaken. The Arksey & O’Malley Framework for conducting scoping studies/reviews has been cited, but it is unclear whether the elements of the framework were followed throughout the conduct and reporting of the scoping review. Lack of clear or detailed reporting of the methods and results of the scoping review component mean that it is difficult to establish the rigor and outcomes of this step in the research. Some limited results of the literature review are provided, but details of e.g. the studies excluded, methods used to select studies and chart data, are not provided - Numbers of participants in the group discussions is not completely clear (e.g. did participants specific discussions participate in other discussions too?). Total number of participants for all three could be provided. Interviews:  - At line 26 (page 7) “the improvement project” is mentioned. Were all participants involved in the same improvement project or multiple projects? What was this project about? Piloting:  - At line 29 (page 8) “10-15 minutes” has been identified as an “acceptable time” for completing the tool. How was this time arrived at? E.g. Was acceptability in terms of time to complete the tool derived from the results obtained from discussions with participants? - It is somewhat unclear whether users filling out the tool used it in relation to the improvement project/s they were themselves involved in. How many projects was the tool piloted for? Some detail regarding the nature of the projects themselves could be useful and helpful for the reader to understand the setting/s where the tool was piloted and how/if this might impact upon using the tool in different settings. Overall, the manuscript is well written and describes the rigorous development of a tool to support sustainability of improvement projects.
--	---

VERSION 1 – AUTHOR RESPONSE

Reviewer Comments and Author Response

We would like to thank the reviewers for their time in reviewing this paper as well as their thoughtful questions and comments. We have now responded to the comments and suggested and updated the manuscript. Our responses have been organised within 4 tables:

Table 1 Reviewer 1: Samuel Silver Page 1-4

Table 2 Reviewer 2: Monica Nyström Page 5-22

Table 3 Reviewer 3: Diane Lorenzetti Page 23-30

Table 4 Reviewer 4: Micah D J Peters Page 31-33

Table 1	
Reviewer 1 Comments Name: Samuel Silver	Author Response
Major Comments: 1. Please provide more information on the pilot testing?  Was training provided to use the tool?  Was the tool piloted on actual QI projects or fictional QI projects? These questions relate to the generalizability of the tool, which should be discussed.	1. Thank you for your questions on piloting.  Training was provided during piloting in the form of a short presentation given to the group on the tool design and instructions for use. The tool questionnaire also included a short introduction and description for use which reads: This tool aims to aid you in planning for long term success of your work. You will be asked to rate 12 factors that have been identified to impact long term success from current literature and evidence. Each rating should represent an overall impression of how you believe your project is doing. Please use the boxes to highlight any comments or actions needed to address the factors.  The tool was piloted on actual QI projects, from diverse settings and intervention types. This information along with example projects are now included on page 11 and 12. V. Small Scale Trailing: CLAHRC NWL fellows (n=11) trialled a draft version of the tool in June 2014. Each fellow was undertaking a QI project across diverse topic areas and settings (for example, service redesign, app development, patient experience measurement and staff training package development). Trialling the tool resulted in refinement the tool's prompt text to reduce the overall length. VI. Piloting: Piloting tool place with 106 participants (83 of which returned a completed tool to the research team). Fifty-two participants indicated that were involved in active QI projects. This included 9 CLAHRC NWL QI projects across diverse topics (such as Sickle Cell Disease, Allergic conditions in children, Polypharmacy in the Elderly, Chronic Obstructive Pulmonary Disease, and Congestive Heart failure) as well as 19 projects outside of the CLAHRC NWL programme.
2. The manuscript requires a discussion of study limitations. Some issues I noticed include:  The tool was developed by people with significant QI experience. Will this tool be generalizable to teams with less QI experience? 	While we were unable to add all the information below to the manuscript we have added a discussion on limitations to page 19.  Thank you for this comment. The process of involving those would use the tool in the design and piloting of the tool was to ensure the tool could be used by all people involved in an improvement project. Tailoring of the language and the instructions were done to ensure people with little QI experience or knowledge would be able to use the tool. Further observation and study of the application of the

 The study did not use Delphi methodology and comments were not anonymous, so how can the authors be certain all opinions were represented (especially since qualitative interviews did not reach saturation and many participants may have had a prior relationship with the interviewer and so restricted their comments)? The tool assumes all sustainably factors are created equal, which is not always the case. Was an attempt made to determine which sustainability factors were most important? 	tool is underway to see if application is impacted by this design. We have noted the need for future research to investigate tool engagement on page 20.  Unfortunately, we cannot ensure all opinions are represented but our study design sought to engage with and take feedback from as many perspectives and individuals as possible. We have recognised the potential bias caused by previous relationships with the participants in the limitations section Page 19 which reads: Another limitation of this work is the potential for responder bias throughout development stages. Prior relationships between researchers and participants was identified as a possible source of bias, namely, social desirability bias, as participants may have responder in ways that were seen as more desirable to the researchers.(47) As the development of the tool was centred on user preferences attempts were made to communicate and reiterate there were no 'right' answer to questions. We also attempted to mitigate this effect by having multiple stages for feedback, with diverse facilitators and a wide variety of participants. We also had a researcher unknown to the majority of the interviewees conduct the interviews.  Thank you for this thoughtful question on the equality of factors. We thought a great deal about the use of weighting or prioritization for the factors prior to designing the tool. Unfortunately, the literature has not reached a consensus on what may be most important to consider for sustainability so we made the conscious decision to avoid weighting and allow teams to score a wide breadth of factors and allow the scores to decide where action should be taken. The tool is now being prospectively studied and this issue is a key research question for this work- What factors appear to have the greatest impact on projects success and when? Studying the application and scores from the tool will provide insight into what teams perceive to be the greatest risks and how they act to mitigate these risks. This may inform future weighting or prioritization of factors.
3. The authors should be more cautious on some of their interpretations, since the real-world efficacy of this tool still needs to be demonstrated. For example, it has yet to be proven that with actual QI projects among less experienced teams the tool is:  Quick to complete Involves simple interpretation Leads to suggestions to address sustainability issues The authors indicate this is the case on page 18, but I think more validation is needed to	Thank you for this observation. These statements were anecdotal observations which will be researched further. We have removed this text to avoid confusion and reinforced that the tool is intended to provide a platform for teams to identify risks and prompt action to enhance sustainability but this will need further research to investigate its use and impact in practice.

support this statement. At the very least, it should be stated that this was the experience in the pilot phase, and whether this will extend to actual QI is unknown.	
4. Future plans for the tool should be discussed. Do the authors intended to validate it prospectively and in different environments? Will the authors evaluate whether use of the tool actually leads to more sustained QI?	Thank you for your interest in our future work. Yes, a prospective longitudinal study has been taking place since the application of the tool in 2015. Teams are currently being followed up 1-1.5 years post funding to understand how they have progressed or sustained. This work will be the focus of a future publication. We have now explicitly discussed this in the Future Research Section in the Discussion page 20. The revised text reads: While attempts have been made to respond to user preferences and create a practical tool, further research is required to assess tool effectiveness and engagement over time. A 3-year programme of research with teams at CLAHRC NWL and other groups internationally is currently underway to investigate tool impact on initiative processes and practices and examine actions taken by improvement teams to sustain improvements across diverse settings and environments. This longitudinal study will also investigate tool links to sustainability outcomes to assess what impact tool use may have on sustained QI projects.
Minor Comments: 1. Abstract: The details of the scoping literature review methodology should be provided. 2. The “design” part of the abstract is not clear. I cannot tell the order in which feedback was elicited. Maybe using numbers would help. 3. Consider mention the roles of all stakeholders in the abstract (e.g., patients, carers, industry, government, etc.) 4. Page 8: Please list the number of the different stakeholder groups involved (e.g., number of patients and carers) 5. Page 11: Text is quite jumbled. Maybe consider presenting the different factors with bullet points or numbers? 6. Page 17, line 24: I think the authors mean to refer to Table 3? 7. Figure 3 does not add much. Please consider removing.	1. Due to word limitation in the abstract we were unable to add this detail but full details on the scoping review now appear in the manuscript methods page 6/7, and results page 9/10 2. Thank you for this feedback. This has now been clarified in the abstract and the methods and results have been reported in numbered stages 1-6. 3. Due to word limit it was not possible to add this information to the abstract but we have included this information in detail within the manuscript in the results section with Table 1 page 9. 4. Table 1 now specifies participant numbers and roles on page 9. 5. Thank you for this comment. The factors have now been numbered 1-12. 6. Table number changed 7. Thank you for the suggestion. The authors agree and have removed this figure.

Table 2	
Reviewer 2 Comments: Name: Monica Nyström	Author Response
1. Is the research question or study objective clearly defined? My main concern is the objective of the study that makes it difficult to separate it from a description of a development process. In the Abstract the objective of the study is described as: "This work aimed to collaborate with stakeholders to develop an evidence-based sustainability tool relevant to people in healthcare settings, and practical for use in improvement initiatives. On page 5 the purpose of the study is described as: "The purpose of this study was to develop a tool for improvement teams to plan, and reflect on, factors important to sustainability at all stages of an initiative, to prompt discussion and action to enhance chances of achieving sustainability." For me this aim/objective/purpose is that of a development project rather than of a scientific study and there need to be something to study that we can learn from and a clear aim/objective/purpose for the study of the development at hand. Some not so elaborated examples that I can think of might clarify what I mean: 1. What factors contributed to x and x during the collaborative design process and how did it change the model? = focus of the study is to provide knowledge on a collaborative design process and what it is that is important during such processes (co-design) 2. What are the differences/similarities between the model extracted from the literature review and the subjective experiences/views of professionals is another option (though the themes might have been studied by others already) = focus of the study is to compare the theoretically derived constructs/themes with users' understanding/experiences of what it is that is important for sustainability of improvement 3. The description of the literature review and how the first model/themes were derived might also act as a background to how the instrument was developed and then the focus is on a pilot study of user's perceived relevance and understanding of the derived concepts (themes) and perceived usability among their	Thank you for this comment. We apologise for the confusion the lack of explicit research questions caused. Your suggestions have picked up on the implicit questions we endeavoured to explore but have now explicitly stated within the introduction of the paper on Page 5-6. While the purpose of the study was to develop a tool, in order to do this, we had to explore how the literature findings matched the experience of those who will use the tool and consider how it could be designed to fit their needs. Our research questions have now been linked to methods and results to reflect our findings on each question. Revised introductory text Page 5-6 reads: The purpose of this study was to collaborate with stakeholders to develop a sustainability tool relevant to people in healthcare settings, and practical for use in improvement initiatives. In order to inform the tool development, we explored the following research questions: 1. How do sustainability factors identified in the literature resonate with the experience of those in improvement projects in healthcare? 2. What features or characteristics make a sustainability tool valuable, practical and useful in real world healthcare practice? Revised methods text Page 6 reads: Methods: Tool development was conducted in 6 stages. The first 3 stages: scoping review, group discussions, and the stakeholder engagement event focussed on reviewing the literature findings and their resonance with stakeholders in this setting. The last 3 stages: interviews, small scale trialling and piloting contributed to designing and testing usability of the tool.

teams (and further studies, as suggested, will be on it's actual use and effect in empirical settings). Then the changes to the initial model will be described etc. (se under Study design below)	
At page 5 you state that 'This article describes work undertaken with healthcare professionals, patients, and researchers to develop a tool to meet the needs of people in improvement initiatives'. Were all these stakeholders involved during the entire process or in specific parts (test of themes, test of form and usability)? In what way were patients involved?	This statement has now been removed and participation reported in individual development stages. We have added a section on participant information which describes improvement team composition and involvement on Page 6: Participants in this study included members of CLAHRC NWL improvement initiative teams and staff. These members come from various backgrounds: multi-disciplinary healthcare practitioners (doctors, nurses, allied healthcare professionals), patients, carers, healthcare managers, directors, analysts and researchers (many participants hold overlapping roles ie. nurse who is also a project manager). Other participants were also included at the engagement event and piloting. Although the majority of attendance is from improvement teams, these event were are open to the public so additional participants included students, fellows, community members and industry partners. Specific participation from these groups in is outlined within each development stage and summarised in the results. We have also included Table 1 within the results section which shows participant involvement across development stages.
2. Is the abstract accurate, balanced and complete? It will need revision of objective (se above) – and based on these changes also under Results and Conclusions.	Abstract has been revised to reflect manuscript changes.
3. Is the study design appropriate to answer the research question? The study design is somewhat confusing (also due the discussion on objective/purpose above) – it can be clarified by describing it as three steps/phases that also will clarify the Results section (both sections described below):  1. Literature review and consolidation into a model/theoretically derived themes on sustainability – Methods used (a more specific description of how themes were chosen) Results = Model 1 themes (content) 2. Design process (interactive and iterative) – test of subjective relevance of Model 1 – Methods used, procedure over time, respondents (why this sample of potential users) Results = Description of discrepancies, changes/adjustments made, and Model 2 themes (content) 3. Pilot test of Model 2 – both content and form Methods used, procedure, respondents (why this sample of potential users) Results = Description 	We apologise for the confusion the design of the study has caused. We have worked to address this issue by defining the research questions as well as the aims as described above. The study design has been reorganised to reflect our research questions. Methods and Results sections have now been numbered in the consecutive development stages 1-6. The methods and results now fall under the following headings.  I. Scoping review II. Group discussions III. Stakeholder engagement event IV. Interviews V. Small scale trialling VI. Piloting We believe this structure clearly outlines the steps taken to develop the tool as well as highlights the changes and findings from each consecutive stage. We have also added the text

of perceived relevance and usability, changes/adjustments made, and Model 3 themes (content)	to reflect this on page 6: Tool development was conducted in 6 stages. The first 3 stages: scoping review, group discussions, and stakeholder engagement event focussed on reviewing the literature findings and their resonance with stakeholders in this setting. The last 3 stages: interviews, small scale trialling and piloting contributed to designing and testing usability of the tool.
Also consider and discuss what you mean by developing and evidencebased instrument - it might not be an appropriate term to use depeing on the influence of the process on the limited themes left in the final tool...	Thank you for this comment. After consideration and considering the connotations and possible confusion caused by the term 'evidence based' we have removed this from the description of the tool. We have now simplified the message to include the aim of developing a relevant and practical tool.
4. Are the methods described sufficiently to allow the study to be repeated? The review started by going through 3 previous reviews (from 2005 and 2012). Criteria for including/excluding the other sources of information (the results of literature review, Appendix A) is less clearly described and some considerations that may inform further work on the paper and providing some argumentation behind the 12 factors included (and why other factors were excluded) are suggested below.	Further details on the review have been added to address this comment on Page 7. No factors or themes were eliminated at this stage. All sustainability constructs identified in the approaches were extracted for thematic analysis. Themes were developed by combining similar or overlapping concepts and removing duplicate but no themes were discounted at this stage. The revised text with this additional information reads: I. Scoping Literature Review: A scoping literature review was undertaken to examine the extent, range and nature of research activity related to sustainability approaches.(26) The research question guiding this review was: 'what approaches have been proposed to assess sustainability in healthcare and what sustainability factors are examined in each method'? Identifying relevant studies: A number of reviews had previously been published to identify factors for sustainability.(3,6,27) These reviews were used as a starting point to identify relevant authors and publications including snowballing of relevant journal articles, reference lists and the PubMed options of 'similar article' and 'cited by-' articles. Selecting studies: We sought approaches (published models, tools, strategies, and frameworks) that identified sustainability factors and themes. Papers that introduced or described a sustainability approach were included. Commentary, posters, protocols, conference

	proceedings, editorials and perspectives were excluded. Charting the data: One author (LL) screened the retrieved papers for inclusion and extracted the data from the articles. Data extraction was independently checked against the full-text articles by the second author (CD). Any differences were discussed and inclusion and exclusion criteria were refined to reflect these discussions. Summarizing the results: All sustainability constructs (factors, questions, criteria etc.) identified in the approaches were extracted for thematic analysis. Aggregate themes were developed by combining similar or overlapping concepts and removing duplicate or redundant labels. Overarching sustainability themes were created using a mapping software.(28)
Data collection and informants A clarification of the samples of informants/respondents is needed and why these were selected in each step – it is difficult to understand. Below some questions that popped up while reading the text. I think you need to provide some text on the participants in the CLAHRC NWL Improvement projects and their role/profession etc., maybe under Setting? Maybe they are both clinical staff and trained to participate in CLAHRC NWL Improvement projects? [ ] Clarify who you refer to as ‘stake holders’ in this case and which method that is used to capture each type of stakeholder’s experience/view. “CLAHRC NWL improvement initiative teams and staff, team members” – clarification regarding what/who you refer to would facilitate the reading experience (Do these improvement initiative teams not include staff and if so which staff? Clarify which staff are you referring to? Clinical staff? Staff at the CLARHRC NWL? Is this the same?)	Thank you for this suggestion. We have now added additional participant information on Page 6 to describe the participants from CLAHRC NWL improvement projects. Members of the project teams are often both healthcare staff and members of the QI project team. We hope this additional text along with the description of participation within Table 1 on Page 9 will clarify this issue. Participants in this study included members of CLAHRC NWL improvement initiative teams and staff. These members come from various backgrounds: multi-disciplinary healthcare practitioners (doctors, nurses, allied healthcare professionals), patients, carers, healthcare managers, directors, analysts and researchers (many participants hold overlapping roles ie. nurse who is also a project manager). Other participants were also included at the engagement event and piloting. Although the majority of attendance is from improvement teams, these event were are open to the public so additional participants included students, fellows, community members and industry partners. Specific participation from these groups in is outlined within each development stage and summarised in the results.
[ ] Are the informants overlapping - participants in the 1) Facilitated group discussion, 2) the Collaborative Learning events, 3) interviews,) the Small scale trialing and 5) Piloting? Clarify... Maybe use a Figure/Table	Many participants participated in multiple stages of development particularly from the stakeholder engagement events and the piloting. As you have suggested, we have now added a Table which

over all the Phases (se above) and each of the steps where data was collected (six steps with the scoping literature review). Why so few clinical staff (if this is the case) – especially when testing the perceived usability?	details participant number and roles (Table 1 page 9). As you will notice in the table, the tool was tested with a significant proportion of healthcare staff at all stages of development.
☒ Why are you reporting sample size: (n=12) in table 1 (COREQ) but a larger number of participants in group-discussions and at the stakeholder event in the Methods section?	Apologies for this error. The previous number reported only on the number of participant in the interviews. This has been amended in the table to reflect participant numbers in each development stage.
The role of the authors/researchers during data collection – active or just observing the discussions? Who led the group discussions, engagement events and small scale trialling?	Researchers within this study acted as participant observers. They provided the introduction and teaching to participants during development stages and provided facilitation during the group discussions and trialling. and A sentence within the methods has been added to clarify the role of researcher throughout this study on Page 6: The researchers within this study had participant observer roles. They provided teaching, facilitation, and explanation throughout the development stages. Linked to this we have also recognised the potential bias caused by the role of the researchers and their previous relationships with the participants in the limitations section Page 19 which reads: Another limitation of this work is the potential for responder bias throughout development stages. Prior relationships between researchers and participants was identified as a possible source of bias, namely, social desirability bias, as participants may have responder in ways that were seen as more desirable to the researchers.(47) As the development of the tool was centred on user preferences attempts were made to communicate and reiterate there were no ‘right’ answer to questions. We also attempted to mitigate this effect by having multiple stages for feedback, with diverse facilitators and a wide variety of participants. We also had a researcher unknown to the majority of the interviewees conduct the interviews.
☐ Sampling of informants/respondents: Regarding the interviews you state that ‘Participants were selected based on their role within the improvement project, their level of knowledge of the project, and their experience with the SM.’ Clarify “improvement projects”	Thank you for noticing this issue. Participants were from multiple CLAHRC NWL projects. This has now been clarified on Page 8 where we also provide rationale for the experience with SM.

(=CLAHRC NWL Improvement projects?) as this can be confusing for a reader. Maybe also elaborate just a little on why their experience of the SM is relevant?	III. Interviews: Interviews aimed to collect in-depth information on value and practicality of tool design. A purposive sampling strategy was used to recruit interviewees. Participants were selected based on their role within diverse CLAHRC NWL improvement projects, their level of knowledge of their project, and their experience with the SM (we sought both those with and without experience in using the SM to ensure we had a balanced sample of those with prior opinions of the SM). This approach aimed to maximize the diversity of perspectives gained from the interviews.(29)
□ Elaborate a little more on the information regarding the interview guide. No questions on content/themes and their relevance at this stage?	The interview guide used open ended questions about tool value and features that would be most or least desirable to see. This was to allow interviewees to identify what they prioritised. While no specific questions on the themes were asked these questions allowed open discussion on whatever the interviewee deemed relevant. This was a deliberate choice as the themes and factors had gone through two iterations and participant comments so further in-depth study of them was deemed unnecessary. The final interview question showed participants an early mock-up of the tool which they commented freely on which included the themes and content.
□ In the methods section you describe a 'Small scale trialing' (n=15) and a 'Pilot' (n=83) of the LTST. Where and with whom were these "tests" conducted? (In ongoing improvement efforts at work place settings? With clinical staff? Managers? Stakeholders? Process supporters?) The information provided regarding these two events leaves the reader with several questions.	Further details on both the small scale testing and piloting have been added to the manuscript. Both the participants and the improvement projects have been elaborated on. Participation from specific groups is outlines in Table 1 on Page 9. The tool was piloted on actual QI projects, from diverse settings and intervention types. This information now appears in the results section along with example projects page 11 and 12. V. Small Scale Trailing: CLAHRC NWL fellows (n=11) trialled a draft version of the tool in June 2014. Each fellow was undertaking a QI project across diverse topic areas and settings (for example, service redesign, app development, patient experience measurement and staff training package development). VI. Piloting: Piloting tool place with 106 participants (83 of which returned a completed tool to the research team). Fifty-two participants indicated that were involved in active QI projects. This included 9 CLAHRC NWL QI projects across diverse topics (such as Sickle Cell Disease, Allergic conditions in children, Polypharmacy in

	the Elderly, Chronic Obstructive Pulmonary Disease and Congestive Heart failure) as well as 19 projects outside of the CLAHRC NWL programme.
Data analysis You state (in table 1 COREQ) that your themes were derived from data. What is the difference between an identified theme and the listed factor from the literature? Are grouped factors = themes? A section describing the data analysis procedure is missing for all types of data collections (except maybe for the literature review) and information in table 1 does not describe all procedures.	Thank you for this question. Themes within the scoping review were developed by combining similar and overlapping constructs from identified sustainability approaches (these were either questions, criteria or factors for sustainability) that were separated by language or label differences. We have now provided information theme creation on Page 7: All sustainability constructs (factors, questions, criteria etc.) identified in the approaches were extracted for thematic analysis. Aggregate themes were developed by combining similar or overlapping concepts and removing duplicate or redundant labels. Overarching sustainability themes were created using a mapping software.(29) Factors were developed through theme consolidation and discussions with participant where adaptation and clarification to the language took place. This has been highlighted on Page 10: II. Group Discussions: In total 22 individuals participated in the internal CLAHRC NWL group discussions. Discussions lead to combining themes that had different labels but were seen as having related or overlapping definitions. Discussions also identified where themes may be confusing and need to be expanded to underlying concepts to be relevant to improvement setting. For example the literature theme of ‘staff skills and capabilities’ was expanded to include skills and capabilities of all those involved which may include as patients, carers or other stakeholders who often participate in QI projects. Academic jargon and terms were also removed such as ‘routinisation’ which were seen as unhelpful or potentially confusing. These discussions resulted in changes to the language used and theme consolidation to form a list of 12 factors impacting sustainability. (Fig 1)
A clearer description of if and how, the model/themes was re-designed during the development process – i.e. the emergence of	Thank you for this comment. We constructed a diagram similar to Fig 1 with more detail on the iteration of the themes and factors but after

new themes and the adjustments of all themes during the process so it is easy to follow the progress of the model/themes – and the arguments/discrepancies that occurred during each phase/step. Figure 1 provides some of this information but this can be clearer.

discussion we believe this is too much information to include in a diagram within the manuscript and would be quite onerous for readers to follow. As the majority of changes were to the presentation and language used to articulate the factors and not underlying concepts behind the 12 factors we have decided to exemplify changes made with more description and examples of changes made during the development (removing jargon, tailoring language and clarifying what is included). Some examples include:

Page 10 Group Discussions:

Discussions also identified where themes may be confusing and need to be expanded to underlying concepts to be relevant to improvement setting. For example the literature theme of 'staff skills and capabilities' was expanded to include skills and capabilities of all those involved which may include as patients, carers or other stakeholders who often participate in QI projects. Academic jargon and terms were also removed such as 'routinisation' which were seen as unhelpful or potentially confusing.

Page 10 Stakeholder engagement event:

The majority of the factors resonated with stakeholders and were recognised as relevant to healthcare settings but in some cases the factor language needed to be adapted to align with stakeholder expertise and understanding. For example the factor, 'Fit with Current Practice' was found to be problematic for participants. Although this factor was meant to convey the importance of interventions being aligned with current practice, many stakeholders mentioned that often improvements must be different from the current ways of working so trying to fit in with 'current practices' would not be desirable or possible. The factor was changed to 'Robust and Adaptable Processes' highlighting the need for interventions with the ability to adapt to local settings. Stakeholders also identified missing concepts and concepts they felt were not clearly represented in the current factors. For example, establishing a shared aim for a project was suggested as an explicit prompt underlying the factor 'commitment to the improvement'.

And Page 11: Small scale Trialling

Stakeholders commented that the tool was a good reminder what to consider for sustainability but suggested changes to some of the language within the tool to remove terms perceived as

	'jargon'. For example in the factor 'Resources in place' original prompt text read: 'I am given sufficient headspace and time to dedicate to the improvement', after discussion the term 'headspace' was removed as it was seen as confusing to some participants. We have also added a statement within the discussion to highlight the changes made to factors and language throughout the development Page 18. This work has shown that the majority of factors from the literature resonated well with stakeholders and were recognised as relevant to healthcare settings. In some cases the literature findings needed to be adapted through changes to the language used to align with stakeholder preferences and understanding. Engaging stakeholder in the design process demonstrated that stakeholders valued clarity, conciseness, and simplicity for tool design with simple data interpretation and visual graphs. Receiving ongoing feedback during the development period from those who will use the tool has allowed us to design an approach that has responded to user needs and has addressed issues with language, length, and practicality along the way.
How were the observations analysed? When needed how were the data sources combined during the analyses?	Observations were analysed in a similar fashion for all stages of development where observations were conducted (CLARHC NWL group Discussions, Stakeholder engagement event, small scale trialling and piloting). The purpose of these observations was to capture learning and suggestions for the development of the tool using quotes and close approximations to the language used by participants where possible. At each stage designated note takers from the research team captured field notes on the conversations which took place including opinions shared and the responses to the discussion questions. Field notes were then collected and transcribed by one researcher. Findings from each stage were discussed among the research team and summarised to inform iterations of language and representation of themes and factors which were adapted and presented at consecutive development stages. We have now clarified this process in more detail within the relevant method sections:

	For example on page 7/8 Stakeholder engagement event: Designated note takers captured key learning and suggestions from the discussions. Field notes were collected and transcribed by one researcher. Findings were summarised and fed back to the research team to inform next steps and tool iteration.
Did the design of group discussions allows for new themes to emerge or were the themes presented and then reacted to – if so what are the risks? = Discuss study and methodological limitations under the Discussion section.	Yes, new themes were specifically sought in group discussions and the stakeholder engagement event. We have added text and an example of the emergence of a new theme to the manuscript: Page 7: In facilitated group discussions, participants provided their views on the resonance of these themes as well as identified any missing themes not seen in the literature. And Page 10: Stakeholders also identified missing concepts and concepts they felt were not clearly represented in the current factors. For example, establishing a shared aim for a project was suggested as an explicit prompt underlying the factor ‘commitment to the improvement’.
5. Are research ethics (e.g. participant consent, ethics approval) addressed appropriately? Ethical considerations are addressed according to the rules in the UK (?) I guess, maybe clarify the role of the researchers during data collection here or in the Methods section. This article type is defined as a research	Thank you for this comment. An ethics statement is provided in the manuscript on page 20. Ethics statement: Ethics approval was not required for this work as it was part of a service evaluation and improvement project. All interviewees provided verbal consent for the recording of the interviews and were informed that all data would be anonymised for publication. As detailed in an above comment a section has been added to clarify the role of researchers throughout this study on Page 6:

study and but I still think that ethical approval is not needed	The researchers within this study had participant observer roles. They provided teaching, facilitation, and explanation throughout the development stages. This has also been addressed in potential limitations of the study Page 19: Another limitation of this work is the potential for responder bias throughout development stages. Prior relationships between researchers and participants was identified as a possible source of bias, namely, social desirability bias, as participants may have responder in ways that were seen as more desirable to the researchers.(47) As the development of the tool was centred on user preferences attempts were made to communicate and reiterate there were no 'right' answer to questions. We also attempted to mitigate this effect by having multiple stages for feedback, with diverse facilitators and a wide variety of participants. We also had a researcher unknown to the majority of the interviewees conduct the interviews.
6. Are the outcomes clearly defined? In the results section (page 11-12) you state that the 12 factors have been organized within 3 areas; People, Practice and Setting, and provide table 3 which describes the factors and the statements for rating and supporting questions included within the tool. Why (and for whom) were the factors organized in these three areas and how and when is table 3 used (e.g. is it part of your analysis or development process or is it for 'practical' use?).	The areas of People, Practice and Setting were natural categories which emerged from organising the 12 factors into similar groups. Although multiple options were available we felt these categories provided a logical grouping of factors for presentation and teaching purposes. Table 3 (now appears as Table 2) provides a description of the factors included in the tool along with a definition, prompt and supporting questions. This table is for practical use and has been constructed for the manuscript. Table 2 provides readers with factor definitions and supporting questions which can be used along with the tool questionnaire and Excel Spreadsheet to use the tool in their own settings and calculate their own scores. This has been clarified on Page 19 Table 4: The Long Term Success tool has been designed on the CLAHRC NWL WISH system. For those who do not have access to this system, the Long Term Success Tool questionnaire form and Excel spreadsheet can be downloaded with this paper. The tool can be used along with Table 2 which provides supporting questions to describe the potential items to consider within each factor. The tool can be used by individuals and teams. Responses can be input into the Excel

	spreadsheet which enables users to produce similar graphs and outputs to ones shown in this paper. The spreadsheet enables 8 possible entry points for a team (up to 20 team members) and will aggregate team data overtime for review and action planning. (Appendices B and C)
7. If statistics are used are they appropriate and described fully? Not relevant	n/a
8. Are the literature up-to-date and appropriate? Depending on changes in purpose/objective additional literature (co-design?) may need to be added.	Thank you for this suggestion. We have not added specific literature on co-design or co-production as these methods come with specific techniques and objectives which were not aligned to our study. For most 'co-design' studies there must be involvement from stakeholders before conception of the idea or improvement which was not the case in this study as the need for an improved sustainability tool was already known from previous evaluation. So although we adhered to some co-design principles we do not want to overstate this in the current manuscript. We have however, added further background studies on page 5 to provide further rationale for this work. Specifically, the need to understand how to influence sustainability to make the most of resources and the difficulties of applying methods and issues identified in past method use have been highlighted. In the current healthcare climate of increasing demands and competing priorities for resources, healthcare planners and stakeholders are increasingly concerned with the long term impact of their investments.(3,10) This has highlighted a need to understand how sustainability of improvement initiatives can be influenced and how specific approaches may help support sustainability.(3,10) Defined procedures for addressing sustainability in improvement initiatives do not exist but many have suggested that sustainability indicators or factors can be used to monitor and influence sustainability over-time.(1,4,12–14) Multiple strategies and approaches such as models and frameworks have been designed to highlight such factors but issues with tool design and content have been identified as barriers to their use in healthcare settings.(10,15–18) Specifically, poorly designed constructs, inadequate coverage of items, and lack clear definitions have impacted application and outcomes in past use.(15–18) Using methods well in practice is a recognised challenge for improvement teams, highlight the need for methods to be designed to be practical for use in real-world healthcare settings. (19–22)
9. Do the results address the research question or objective? They describe the development process and the	We have now reorganised the manuscript to ensure the results are aligned with research questions.

resulting tool – otherwise see comments on objective/purpose.	Research questions text on Page 5-6 reads: In order to inform the tool development we explored the following research questions: 1. How do sustainability factors identified in the literature resonate with the experience of those in improvement projects in healthcare? 2. What features or characteristics make a sustainability tool valuable, practical and useful in real world healthcare practice? Revised methods outline how research question 1 will be answered by the first three development stages and Question 2 will be answered by the last 3 stages Page 6: Methods: Tool development was conducted in 6 stages. The first 3 stages: scoping review, group discussions, and the stakeholder engagement event focussed on reviewing the literature findings and their resonance with stakeholders in this setting. The last 3 stages: interviews, small scale trialling and piloting contributed to designing and testing usability of the tool. Our results on Page 9 reflect this: Each step in the methods allowed for iterative development of concepts, content, and design of the tool. Key iterations of the themes, factors, and tool design elements are summarized in Figure 1 and the number and composition of participants is outlined in table 1. The following section discusses findings from each development stage and concludes with an introduction to the resultant Tool.
10. Are they presented clearly? Depends on the more specific objective/purpose. I suggest a presentation in relation to Phases: 1) Literature review and consolidation into a model/theoretically derived themes on sustainability; 2) Design process (interactive and iterative); and 3) Pilot test of Model 2 – both content and form where the LSTS (Model 3) can be presented (and intended use of the LSTS tool (Model 3) is discussed in The Discussion section along with the changes during the design and piloting process and potential benefits and shortcomings this might have added in relation to the first model).	Thank you for this comment. We have taken on board your suggestions and organised the results to reflect the iterative development process stages 1-6. The methods and results now fall under the following headings.  VII. Scoping review, VIII. Group discussions IX. Stakeholder engagement event X. Interviews XI. Small scale trialling XII. Piloting As mentioned in an above comment we believe this structure clearly outlines the methods taken to develop the tool as well as highlights the changes and findings from each consecutive stage.
11. Are the discussion and conclusions justified by the results Discussion I also miss a reflection on how the resulting tool is similar/different from earlier theories/models and tools. For example, CFIR has toolkits that perhaps could be used in practical settings.	Thank you for this comment. We have now added some discussion on how the tool aligns well with other tools as they have similar factors and approaches. We have also highlighted the unique attributes of the Tool which are not found in others as well as the possibility for the use of complimentary tools. (CFIR toolkits have not been referenced here as we could not locate any that pertained specifically to sustainability) Page

	18/19. The LTST builds on established literature and aligns well with other sustainability models and frameworks with all LTST factors reflected in one or more of the other approaches.(1,2,4,9,24,35–44) LTST is distinguished from other approaches by its practical design and ability to draw on team suggestions for action planning. Using participant ideas as a platform for action is a unique feature of the tool that is not present in other tools currently used in this area. Also unique to the LTST is that the allotted time for use, a identified barrier and challenge to other method use, has been explicitly tested and informed by end-users.(8,45) While many other methods involve either unknown or substantial time commitments, the LTST can be completed in approximately 10-15 minutes.(42,45) There is also potential to supplement the use of other models or frameworks to complement the LTST. For example if a project receives a particularly low rating for the factor ‘Robust and Adaptable Processes’, The Model for Highly Adoptable Improvement toolkit by Hayes et al. may be used to aid the team in further understanding of where the intervention can be adapted.(46)
Also, how much did the tool develop from the scoping review to the final tool? What came out from the different steps – and a discussion of the steps would help.	We hope some of our above comments have addressed this question. We believe that now the results are reported more clearly and in consecutive stages the changes made at each will be much more obvious to the reader. Figure 1 also provides a summary of these changes. As mentioned in an above comment to exemplify changes made further description and examples of changes made during the development stages have been included: Page 10 Group Discussions: Discussions also identified where themes may be confusing and need to be expanded to underlying concepts to be relevant to improvement setting. For example the literature theme of ‘staff skills and capabilities’ was expanded to include skills and capabilities of all those involved which may include as patients, carers or other stakeholders who often participate in QI projects. Academic jargon and terms were also removed such as ‘routinisation’ which were seen as unhelpful or potentially confusing. Page 10 Stakeholder engagement event:

	The majority of the factors resonated with stakeholders and were recognised as relevant to healthcare settings but in some cases the factor language needed to be adapted to align with stakeholder expertise and understanding. For example the factor, 'Fit with Current Practice' was found to be problematic for participants. Although this factor was meant to convey the importance of interventions being aligned with current practice, many stakeholders mentioned that often improvements must be different from the current ways of working so trying to fit in with 'current practices' would not be desirable or possible. The factor was changed to 'Robust and Adaptable Processes' highlighting the need for interventions with the ability to adapt to local settings. Stakeholders also identified missing concepts and concepts they felt were not clearly represented in the current factors. For example, establishing a shared aim for a project was suggested as an explicit prompt underlying the factor 'commitment to the improvement'. And Page 11: Small scale Trialling Stakeholders commented that the tool was a good reminder what to consider for sustainability but suggested changes to some of the language within the tool to remove terms perceived as 'jargon'. For example in the factor 'Resources in place' original prompt text read: 'I am given sufficient headspace and time to dedicate to the improvement', after discussion the term 'headspace' was removed as it was seen as confusing to some participants. We have also added a statement within the discussion to highlight the changes made throughout development Page 18. This work has shown that the majority of factors from the literature resonated well with stakeholders and were recognised as relevant to healthcare settings. In some cases the literature findings needed to be adapted through changes to the language used to align with stakeholder preferences and understanding. Engaging stakeholder in the design process demonstrated that stakeholders valued clarity, conciseness, and simplicity for tool design with simple data interpretation and visual graphs. Receiving ongoing feedback during the development period from those who will use the tool has allowed us to design an approach that has responded to user needs and has addressed issues with language, length, and practicality along the way.
A lot of valuable information is provided in this section regarding your thoughts on when and by whom the LTST is supposed to be used. As a reader I would prefer an short introduction to your thoughts on this matter earlier in the manuscript,	We have now added background information to describe issues with current methods for sustainability. We have also expanded information on the CLAHRC Programme and the rationale for a new sustainability tool.

that is elaborate a bit more on the difficulties using the tools available and what they might be lacking.	Page 5: Defined procedures for addressing sustainability in improvement initiatives do not exist but many have suggested that sustainability indicators or factors can be used to monitor and influence sustainability over-time.(1,4,12–14) Multiple strategies and approaches such as models and frameworks have been designed to highlight such factors but issues with tool design and content have been identified as barriers to their use in healthcare settings.(10,15–18) Specifically, poorly designed constructs, inadequate coverage of items, and lack clear definitions have impacted application and outcomes in past use.(15–18) Using methods well in practice is a recognised challenge for improvement teams, highlight the need for methods to be designed to be practical for use in real-world healthcare settings.(19–22) The application of one sustainability method, the NHS Institute for Innovation and Improvement Sustainability Model (SM), has been previously described.(8,23) The SM is a self-assessment tool that details key factors that increase the likelihood of sustainability and continuous improvement.(24) The model is used to raise awareness of key factors for sustainability, and prompt teams to consider actions to increase the likelihood of sustainability.(24) Application of this model was described and results demonstrated that while the SM raised awareness of determinants of sustainability and was perceived as valuable, teams found it difficult to understand and to apply the model routinely.(8,23) In particular, concerns were raised about the clarity the language used within the model, the user-friendliness of design, the length of time taken to complete the questions and suitability for continuous use in healthcare settings.(8) Page 6: Setting: To support multidisciplinary teams to implement changes CLAHRC NWL systematically applies Quality Improvement (QI) methods such as the Model for Improvement and Action Effect Method.(19,23) The approach previously included use of the SM (2008-2013) but following internal evaluation and published research, it was acknowledged that a new more user-friendly method for sustainability was required to meet the needs of improvement teams.(8,23)
Methodological considerations/study limitations are not addressed adequately in this section. Conclusion section Needs to follow a clarified objective/purpose of the study.	A limitations section has now been added on Page 19: Limitations: A limitation of this work is the use of a snowballing scoping review opposed to a systematic review. Conducting a full systematic

12. Are the study limitations discussed adequately? Study limitations have not been addressed – this needs to be added.	review may have uncovered further articles and/or approaches but due to the practical time constraints of our programme this was not possible. The results of our review have fed into a protocol for a full systematic review on available sustainability approaches which is now underway.(47) The results of this review will inform future adaptation of the LTST. Another limitation of this work is the potential for responder bias throughout development stages. Prior relationships between researchers and participants was identified as a possible source of bias, namely, social desirability bias, as participants may have responder in ways that were seen as more desirable to the researchers.(48) As the development of the tool was centred on user preferences attempts were made to communicate and reiterate there were no 'right' answer to questions. We also attempted to mitigate this effect by having multiple stages for feedback, with diverse facilitators and a wide variety of participants. We also had a researcher unknown to the majority of the interviewees conduct the interviews.
---	--

Table 3	
Reviewer 3 Comments: Name: Diane Lorenzetti	Author Response
This paper describes the development of a stakeholder-driven tool to assess the ongoing and long-term sustainability of healthcare improvement projects and programs. As the sustainability of new initiatives is often a barrier to both implementation and ongoing support, a tool	Thank you for your thoughtful review and helpful suggestions. We respond to each below.

to aid decision makers and other stakeholders to conduct formative assessments of such initiatives is an important timely contribution to both practice, and the existing literature on this subject. BACKGROUND I appreciate the authors clearly articulating the definition of sustainability that guided their project and tool development. On page 5 the authors refer to a sustainability model developed by the NHS Institute for innovation and improvement sustainability. While the authors state that this model has been reported elsewhere, I would suggest it is important to include a brief description of this model here as well, particularly as it appears that this work may have at least partially inspired the present study.	As suggested extra detail on the SM has now been added to provide detail on the model and its aims. The revised text appears page 5. The application of one sustainability method, the NHS Institute for Innovation and Improvement Sustainability Model (SM), has been previously described.(8,23) The SM is a self-assessment tool that details key factors that increase the likelihood of sustainability and continuous improvement.(24) The model is used to raise awareness of key factors for sustainability, and prompt teams to consider actions to increase the likelihood of sustainability.(24) Application of this model demonstrated that while the SM raised awareness of determinants of sustainability and was perceived as valuable, teams found it difficult to understand and to apply the model routinely.(8,23) In particular, concerns were raised about the clarity the language used within the model, the user-friendliness of design, the length of time taken to complete the questions and suitability for continuous use in healthcare settings.(8)
While I realize that the scoping review the authors have undertaken was meant to identify existing research on program sustainability, the inclusion of a few additional studies in the background section of this paper that speak to existing gaps in the literature would strengthen the rational for this work.	Thank you for this suggestion, we have now added further background studies on page 5 to provide further rationale for this work. Specifically, the need to understand how to influence sustainability to make the most of resources and the difficulties of applying methods and issues identified in past method use have been highlighted. In the current healthcare climate of increasing demands and competing priorities for resources, healthcare planners and stakeholders are increasingly concerned with the long term impact of their investments.(3,10) This has highlighted a need to understand how sustainability of improvement initiatives can be influenced and how specific approaches may help support sustainability.(3,10) Defined procedures for addressing sustainability in improvement initiatives do not exist but many have suggested that sustainability indicators or factors can be used to monitor and influence sustainability over-time.(1,4,12–14) Multiple strategies and approaches such as models and

	frameworks have been designed to highlight such factors but issues with tool design and content have been identified as barriers to their use in healthcare settings.(10,15–18) Specifically, poorly designed constructs, inadequate coverage of items, and lack clear definitions have impacted application and outcomes in past use.(15–18) Using methods well in practice is a recognised challenge for improvement teams, highlight the need for methods to be designed to be practical for use in real-world healthcare settings. (19–22)
METHODS Scoping Review The authors have not included a detailed methodology for their scoping review. The Arksey and O’Malley framework adopted for this review outlines specific steps for conducting a comprehensive and transparent review. This framework includes: “identifying the research question, searching for relevant studies, selecting studies, charting the data, and collating, summarizing and reporting the results, and consulting with stakeholders to inform or validate study findings”. While no commonly agreed guidelines exist for the reporting of scoping reviews, authors often incorporate a level of detail in their reporting that is similar to that outlined in the PRISMA Guidelines for the Reporting of Systematic Reviews (http://prisma-statement.org/). This includes: eligibility criteria, a list of information sources, a detailed search strategy (or list of search terms used) and any search limits such as language or date, study selection process (which often incorporates dual review), and the charting process/thematic analysis that was adopted (including a partial list of data elements that were extracted). While some of these elements are present in this manuscript, I would recommend the authors include additional details on their approach.	Thank you for this comment. We have now added further details on our scoping review and written this section in a more structured format with sub-headings to aid in clarity for readers. Page 6/7. The revised text reads: A scoping literature review was undertaken to examine the extent, range and nature of research activity related to sustainability approaches.(26) The research question guiding this review was: ‘what approaches have been proposed to assess sustainability in healthcare and what sustainability factors are examined in each method’? Identifying relevant studies: A number of reviews had previously been published to identify factors for sustainability.(3,6,27) These reviews were used as a starting point to identify relevant authors and publications including snowballing of relevant journal articles, reference lists and the PubMed options of ‘similar article’ and ‘cited by-’ articles. Selecting studies: We sought approaches (published models, tools, strategies, and frameworks) that identified sustainability factors and themes. Papers that introduced or described a sustainability approach were included. Commentary, posters, protocols, conference proceedings, editorials and perspectives were excluded. Charting the data: One author (LL) screened the retrieved papers for inclusion and extracted the data from the articles. Data extraction was independently checked against the full-text articles by the second author (CD). Any differences were discussed and inclusion and exclusion criteria were refined to reflect these discussions. Summarizing the results: All sustainability constructs (factors, questions, criteria etc.) identified in the approaches were extracted for thematic analysis. Aggregate themes were developed by combining similar or overlapping

In their 2005 article, Arksey and O'Malley stress the need to be "as comprehensive as possible in identifying primary studies (published and unpublished) and reviews suitable for answering the central research question". While consulting prior reviews, searching reference lists, and citation searching/related articles in PubMed are all useful approaches to study identification, comprehensive searching of relevant electronic databases is essential to conducting a scoping review. I can't tell if the authors constructed detailed search strategies to search PubMed, or relied on reference lists and snowball sampling of studies included in prior reviews. A detailed search of PubMed and other databases (such as EMBASE or CINAHL) might have yielded important frameworks and models that could have informed tool development. While I am not suggesting the authors expand their search, I do feel that it is important to provide additional details on the search process and address any search limitations in the Discussion section of this manuscript.	concepts and removing duplicate or redundant labels. Overarching sustainability themes were created using a mapping software.(28) Thank you for your question regarding the database searches. A preliminary database search using key words (sustainability, methods, tools, and approaches) was attempted. This search received over 39 000 publications. The first 100 titles and abstracts were screened by authors and returned few relevant papers. Considering the ambiguity of the search terms and after discussion with co-authors and a medical librarian, a snowballing and citation tracking scoping review was suggested, given the time constraints of our programme. We recognise this approach may have resulted in missed publications so we have now added this as a specific limitation of the work and reported on the conduct of a future systematic review on the topic. On page 20 we report: A limitation of this work is the use of a snowballing scoping review opposed to a systematic review. Conducting a full systematic review may have uncovered further articles and/or approaches but due to the practical time constraints of our programme this was not possible. The results of our review have fed into a protocol for a full systematic review on available sustainability approaches which is now underway.(46) The results of this review will inform future adaptation of the LTST.
1.Group Discussions, Stakeholder engagement and Interviews. Could the authors comment, in their manuscript, on prior relationships, if any, researchers had with focus group and interview participants?	The researchers were known to and had prior relationships with the participants in the CLAHRC NWL discussion groups. The researcher conducting the interviews had no prior relationship with any of the interviewees besides the 2 QI facilitators who worked with teams but were employed within CLAHRC NWL. We have now provided information on researcher relationships with participants and commented on this in the limitation section page 20. Another limitation of this work is the potential for

	responder bias throughout development stages. Prior relationships between researchers and participants was identified as a possible source of bias, namely, social desirability bias, as participants may have responder in ways that were seen as more desirable to the researchers.(47) As the development of the tool was centred on user preferences attempts were made to communicate and reiterate there were no 'right' answer to questions. We also attempted to mitigate this effect by having multiple stages for feedback, with diverse facilitators and a wide variety of participants. We also had a researcher unknown to the majority of the interviewees conduct the interviews.
The authors state that they adopted a thematic content analysis approach (page 7 line 42) to derive themes from their interview data. Can the authors please outline this process in more detail?	Thank you for your interest in the thematic analysis. In order to conduct our thematic analysis, a coding structure based on our interview questions was developed. This structure allowed us to summarise participant responses and form themes within questions on value of specific characteristics, design elements and potential outputs. The coding structure and nodes were iteratively adapted and refined as further interviews were added to the dataset. During this process notes summarising the findings were documented in memos with explicit links to interview text. We have now added some further detail on our process of thematic analysis within the manuscript page 8. Interviews were audio recorded and uploaded onto qualitative software Nvivo (version 9). Audio recordings were coded directly on Nvivo using thematic content analysis.(30) A preliminary coding structure was developed using the interview questions as coding nodes, with themes inductively derived to summarise responses and record patterns in the data. The coding structure was iteratively developed and refined as further interviews were added to the dataset.(31) Results have been summarized using descriptive summaries and example quotes with explicit links to source text.
Trials: Small Scale and Pilot Who were the individuals who participated in the small scale trial and pilot of the tool? Were they program developers, project coordinators, or others with relevant experience?	Thank you for this request. We have now added information about number of participants and participant roles within the results section in Table 1 on page 9. The tool was tested and piloted with diverse individuals who were involved in QI projects. This information along with example projects are now included on page 11 and 12. V. Small Scale Trailing: CLAHRC NWL fellows (n=11) trialled a draft version of the tool in June 2014. Each fellow was undertaking a QI project across diverse topic areas and settings

Can the authors please provide additional detail on how these trials and pilots were conducted? I'm left wondering if participants were all scoring a standardized exemplar project/program or different ones, and how the type of program/project assessed may have influenced/or not influenced the feedback received on the utility of this tool?	(for example, service redesign, app development, patient experience measurement and staff training package development). Trialling the tool resulted in refinement the tool's prompt text to reduce the overall length. VI. Piloting: Piloting tool place with 106 participants(83 of which returned a completed tool to the research team). Fifty-two participants indicated that were involved in active QI projects. This included 9 CLAHRC NWL QI projects across diverse topics (such as Sickle Cell Disease, Allergic conditions in children, Polypharmacy in the Elderly, Chronic Obstructive Pulmonary Disease, and Congestive Heart failure) as well as 19 projects outside of the CLAHRC NWL programme. Participants were asked to score their own QI projects throughout development stages. This has been clarified throughout the methods sections on page 8 and 9: IV. Small scale trialling: A group of individuals involved leading QI projects as part of a CLAHRC NWL fellowship programme were asked to trial a draft version of the tool. Trialling with this group aimed to understand the practical application of the tool including the approximate amount of time to complete by a wide range of people with diverse experience and expertise in improvement initiatives. Each participant filled out the tool for their own QI project. After completion, the group discussed the experience and posed questions on use. Critical feedback and suggestions for tool development were recorded as observation notes and summarised by the research team to inform tool iterations and piloting. V. Piloting: The resulting tool was piloted in July 2014. Piloting aimed to provide an opportunity for further comments and suggestions on practicality of the tool in healthcare settings, and to measure if the tool could be completed within an acceptable timeframe. A brief presentation given to participants to outline the tool design and instructions for use. Participants were asked to fill out the tool for their individual QI projects. Individuals without a formal project were asked to fill out the tool with a hypothetical project in mind.
RESULTS The authors should include a statement outlining the number of studies identified and number of studies included in their scoping review. The authors employed a variety of steps and	Thank you for this comment. We have now added this statement to page 9 which reads: The scoping review identified 81 publications with 35 articles retrieved in full text for full documentary analysis. In total 16 publications

inputs to develop their tool. I appreciate the inclusion of a figure summarizing this process, and the changes that occurred to the tool throughout. On page 9 line 8 the authors state that “these discussions resulted in a consolidated list of 12 factors impacting sustainability”. Suggest citing Figure 1 here. Did these 12 factors change as a result of the stakeholder engagement event and interviews? A statement either way would be helpful.	which identified sustainability approaches were included in this review. Fig 1 has been cited here as suggested. Page 10. The underlying concepts behind the 12 factors did not change but the presentation and language used to articulate the factors was developed and adapted to improve ease of understanding, and user-friendliness. These changes have now been exemplified with more examples of changes at each stage (removing jargon, adjusting language and clarifying what is included). For example revised text within the stakeholder engagement event on Page 10 reads: The majority of the factors resonated with stakeholders and were recognised as relevant to healthcare settings but in some cases the factor language needed to be adapted to align with stakeholder expertise and understanding. For example the factor, ‘Fit with Current Practice’ was found to be problematic for participants. Although this factor was meant to convey the importance of interventions being aligned with current practice, many stakeholders mentioned that often improvements must be different from the current ways of working so trying to fit in with ‘current practices’ would not be desirable or possible. The factor was changed to ‘Robust and Adaptable Processes’ highlighting the need for interventions with the ability to adapt to local settings. Stakeholders also identified missing concepts and concepts they felt were not clearly represented in the current factors. For example, establishing a shared aim for a project was suggested as an explicit prompt underlying the factor ‘commitment to the improvement’.
DISCUSSION The discussion should begin with a broad statement outlining what the authors accomplished in their study and any unique findings that were noted during the process of tool development.	Thank you for this suggestion. We have now added the following text to outline the accomplishment and findings of our study as an introduction to our discussion. Page 18 The aim of this work was to develop a relevant and practical tool for sustainability that meets the needs of people in improvement initiatives. We explored how sustainability factors identified in

	the literature resonated with those in improvement projects and the features or characteristics which make a sustainability tool most valuable in real world healthcare practice. This work has shown that the majority of factors from the literature resonated well with stakeholders and were recognised as relevant to healthcare settings. In some cases the literature findings needed to be adapted through changes to the language used to align with stakeholder preferences and understanding. Engaging stakeholder in the design process demonstrated that stakeholders valued clarity, conciseness, and simplicity for tool design with simple data interpretation and visual graphs. Receiving ongoing feedback during the development period from those who will use the tool has allowed us to design an approach that has responded to user needs and has addressed issues with language, length, and practicality along the way.
The tool the authors have developed could potentially also inform program/project planning efforts. The authors may wish to comment on this in their discussion. Do the authors themselves have specific plans to test their tool in practice and over time?	Thank you for this suggestion. We agree the tool may have application in planning for programmes. This requires further research to understand potential opportunities and risks. Unfortunately we are unable to comment on this in the manuscript due to work limit but we have however discussed plans for future research on Page 19. Future Research: While attempts have been made to respond to user preferences and create a practical tool, further research is required to assess tool effectiveness and engagement over time. A 3-year programme of research with teams at CLAHRC NWL and other groups internationally is currently underway to investigate tool impact on initiative processes and practices and examine actions taken by improvement teams to sustain improvements across diverse settings and environments. This longitudinal study will also investigate tool links to sustainability outcomes to assess what impact tool use may have on sustained QI projects.
OTHER On page 24, the header for Figure 1 is mislabeled as Figure 2. Table 1 consolidated criteria: In the Domain 2 (study design) section of this table, the authors should report on sample sizes and data collection methods for all stages of their study – not just interviews.	Figure number has been changed COREQ table updated with all participants.

Table 4	
Reviewer 4 Comments: Name: Micah D J Peters	Author Response
Introduction and objectives:  - The introduction and objectives section clearly sets up the problem describing how there are significant challenges to the sustainability of healthcare improvement projects. Some limitations around researching sustainability are provided, for example issues with consistency in terms of key definitions and reporting. Key domains of sustainability are briefly introduced. An operating definition of sustainability is provided. Frameworks/models of implementation are introduced and some of the challenged teams face in using them in practice are briefly presented. The objectives of the project/paper are clearly explained and the parameters around the objectives provided. 	Thank you for your review and helpful suggestions. We respond to your suggestions below.
Minor suggestion: replace “patients” with “healthcare consumers”	Thank you for this suggestion. For the purposes of our work and within our setting we believe ‘patients’ reflects the more commonly recognised terminology.
Design: Scoping Literature Review:  - A scoping literature review was undertaken. The Arksey & O’Malley Framework for conducting scoping studies/reviews has been cited, but it is unclear whether the elements of the framework were followed throughout the conduct and reporting of the scoping review. Lack of clear or detailed reporting of the methods and results of the scoping review component mean that it is difficult to establish the rigor and outcomes of this step in the research. Some limited results of the literature review are provided, but details of e.g. the studies excluded, methods used to select studies and chart data, are not provided 	We have now added further details on our scoping review and written this section in a more structured format with sub-headings to aid in clarity for readers. Page 6/7. The revised text reads: A scoping literature review was undertaken to examine the extent, range and nature of research activity related to sustainability approaches.(26) The research question guiding this review was: ‘what approaches have been proposed to assess sustainability in healthcare and what sustainability factors are examined in each method’? Identifying relevant studies: A number of reviews had previously been published to identify factors for sustainability.(3,6,27) These reviews were used as a starting point to identify relevant authors and publications including snowballing of relevant journal articles, reference lists and the PubMed options of ‘similar article’

	and 'cited by-' articles. Selecting studies: We sought approaches (published models, tools, strategies, and frameworks) that identified sustainability factors and themes. Papers that introduced or described a sustainability approach were included. Commentary, posters, protocols, conference proceedings, editorials and perspectives were excluded. Charting the data: One author (LL) screened the retrieved papers for inclusion and extracted the data from the articles. Data extraction was independently checked against the full-text articles by the second author (CD). Any differences were discussed and inclusion and exclusion criteria were refined to reflect these discussions. Summarizing the results: All sustainability constructs (factors, questions, criteria etc.) identified in the approaches were extracted for thematic analysis. Aggregate themes were developed by combining similar or overlapping concepts and removing duplicate or redundant labels. Overarching sustainability themes were created using a mapping software.(28)
- Numbers of participants in the group discussions is not completely clear (e.g. did participants specific discussions participate in other discussions too?). Total number of participants for all three could be provided.	Yes there were a number of participant that participated in more than one discussion. The total number of participant is now noted in the manuscript on Page 10 and within Table 1. II. Group Discussions: In total 22 individuals participated in the internal CLAHRC NWL group discussions. Discussions lead to combining themes that had different labels but were seen as having related or overlapping definitions. Discussions also identified where themes may be confusing and need to be expanded to underlying concepts to be relevant to improvement setting.
Interviews: - At line 26 (page 7) "the improvement project" is mentioned. Were all participants involved in the same improvement project or multiple projects? What was this project about?	Thank you for noticing this issue. Participants were from multiple CLAHRC NWL projects. This has now been clarified on Page 8. III. Interviews: Interviews aimed to collect in-depth information on value and practicality of tool design. A purposive sampling strategy was used to recruit interviewees. Participants were selected based on their role within diverse CLAHRC NWL improvement projects, their level of knowledge of their project, and their experience with the SM (we sought both those with and without experience in using the SM to ensure we had a balanced sample of those with prior opinions of the SM). This approach aimed to maximize the diversity of perspectives gained from the interviews.(29)
Piloting: - At line 29 (page 8) "10-15 minutes" has been identified as an "acceptable time" for	Thank you for pointing out this error. This time frame belongs in the results section as it resulted from the small scale testing with fellows who

completing the tool. How was this time arrived at? E.g. Was acceptability in terms of time to complete the tool derived from the results obtained from discussions with participants?	were able to complete the tool within 10-15 minutes and stated that this was an acceptable time frame as practitioners and healthcare users would not have much more time to dedicate to the tool. We have now deleted this text from the methods and have only discussed this within the results on Page 11: All participants completed the tool within 15 minutes. This timeframe was discussed and seen as acceptable, with the fellows commenting that no more than 15 minutes should be allotted for routine tool use in practice.
-It is somewhat unclear whether users filling out the tool used it in relation to the improvement project/s they were themselves involved in. How many projects was the tool piloted for? Some detail regarding the nature of the projects themselves could be useful and helpful for the reader to understand the setting/s where the tool was piloted and how/if this might impact upon using the tool in different settings. Overall, the manuscript is well written and describes the rigorous development of a tool to support sustainability of improvement projects.	The tool was piloted on actual QI projects, from diverse settings and intervention types. This information along with example projects are now included on page 11 and 12. V. Small Scale Trailing: CLAHRC NWL fellows (n=11) trialled a draft version of the tool in June 2014. Each fellow was undertaking a QI project across diverse topic areas and settings (for example, service redesign, app development, patient experience measurement and staff training package development). VI. Piloting: Piloting tool place with 106 participants. Fifty-two participants indicated that were involved in active QI projects. This included 9 CLAHRC NWL QI projects across diverse topics (such as Sickle Cell Disease, Allergic conditions in children, Polypharmacy in the Elderly, Chronic Obstructive Pulmonary Disease and Congestive Heart failure) as well as 19 projects outside of the CLAHRC NWL programme.